# Uncovering circuit mechanisms of current sinks and sources with biophysical simulations of primary visual cortex

Atle E Rimehaug[1]*, Alexander J Stasik[2], Espen Hagen[2,3], Yazan N Billeh[4], Josh H Siegle[4], Kael Dai[4], Shawn R Olsen[4], Christof Koch[4], Gaute T Einevoll[2,5], Anton Arkhipov[4]*[†]

[1]Department of Informatics, University of Oslo, Oslo, Norway; [2]Department of Physics, University of Oslo, Oslo, Norway; [3]Department of Data Science, Norwegian University of Life Sciences, Ås, Norway; [4]MindScope Program, Allen Institute, Seattle, United States; [5]Department of Physics, Norwegian University of Life Sciences, Ås, Norway

*For correspondence:
atleeri@ifi.uio.no (AER);
antona@alleninstitute.org (AA)

[†]Lead contact

**Abstract** Local field potential (LFP) recordings reflect the dynamics of the current source density (CSD) in brain tissue. The synaptic, cellular, and circuit contributions to current sinks and sources are ill-understood. We investigated these in mouse primary visual cortex using public Neuropixels recordings and a detailed circuit model based on simulating the Hodgkin–Huxley dynamics of >50,000 neurons belonging to 17 cell types. The model simultaneously captured spiking and CSD responses and demonstrated a two-way dissociation: firing rates are altered with minor effects on the CSD pattern by adjusting synaptic weights, and CSD is altered with minor effects on firing rates by adjusting synaptic placement on the dendrites. We describe how thalamocortical inputs and recurrent connections sculpt specific sinks and sources early in the visual response, whereas cortical feedback crucially alters them in later stages. These results establish quantitative links between macroscopic brain measurements (LFP/CSD) and microscopic biophysics-based understanding of neuron dynamics and show that CSD analysis provides powerful constraints for modeling beyond those from considering spikes.

## Editor's evaluation

The study demonstrates that utilizing the LFP and/or the CSD in modeling can facilitate model configuration and implementation by revealing discrepancies between models and experiments. The analysis of the biophysical origin of the canonical CSD using the model is an interesting and worthy line of investigation. The dissection of CSD components is detailed and exhaustive. A key novelty of this article is the addition of CSD patterns as another constraint to more accurately infer the model parameters beyond its prior state.

## Introduction

The local field potential (LFP) is the low-frequency component (below a few hundred Hertz) of the extracellular potential recorded in brain tissue that originates from transmembrane currents in the vicinity of the recording electrode (*Lindén et al., 2011*; *Buzsáki et al., 2012*; *Einevoll et al., 2013*; *Pesaran et al., 2018*; *Sinha and Narayanan, 2022*). While the high-frequency component of the extracellular potential, the single- or multi-unit activity (MUA), primarily reflects action potentials of one or more nearby neurons, the LFP predominantly stems from currents caused by synaptic inputs

(*Mitzdorf, 1985*; *Einevoll et al., 2007*) and their associated return currents through the membranes. Thus, cortical LFPs represent aspects of neural activity that are complementary to those reflected in spikes, and as such, they can provide additional information about the underlying circuit dynamics from extracellular recordings.

Applications of LFP are diverse and include investigations of sensory processing (*Rall and Shepherd, 1968*; *Di et al., 1990*; *Victor et al., 1994*; *Kandel and Buzsáki, 1997*; *Mehta et al., 2000a*; *Mehta et al., 2000b*; *Henrie and Shapley, 2005*; *Einevoll et al., 2007*; *Belitski et al., 2008*; *Montemurro et al., 2008*; *Niell and Stryker, 2008*; *Nauhaus et al., 2009*; *Bastos et al., 2015*; *Senzai et al., 2019*), motor planning (*Scherberger et al., 2005*; *Roux et al., 2006*), navigation (*Tort et al., 2008*; *Makarova et al., 2011*; *Fernández-Ruiz et al., 2012*; *Watrous et al., 2013*; *Fernández-Ruiz et al., 2017*), and higher cognitive processes (*Pesaran et al., 2002*; *Womelsdorf et al., 2006*; *Liu and Newsome, 2006*; *Kreiman et al., 2006*; *Liebe et al., 2012*). The LFP is also a promising signal for steering neuroprosthetic devices (*Mehring et al., 2003*; *Andersen et al., 2004*; *Rickert et al., 2005*; *Markowitz et al., 2011*; *Stavisky et al., 2015*) and for monitoring neural activity in human recordings (*Mukamel and Fried, 2012*) because the LFP is more easily and stably recorded in chronic settings than spikes. Due to the vast number of neurons and multiple neural processes contributing to the LFP, however, it can be challenging to interpret (*Buzsáki et al., 2012*; *Einevoll et al., 2013*; *Hagen et al., 2016*). While we have extensive phenomenological understanding of the LFP, less is known about how different cell and synapse types and connection patterns contribute to the LFP or how these contributions are sculpted by different information processing streams (e.g., feedforward vs. feedback) and brain states.

One way to improve its interpretability is to calculate the current source density (CSD) from the LFP, which is a more localized measure of activity, and easier to read in terms of the underlying neural processes. The current sinks and sources indicate where positive ions flow into and out of cells, respectively, and are constrained by Kirchoff's current law (i.e., currents sum to zero over the total membrane area of a neuron). However, the interpretation of current sinks and sources is inherently ambiguous as several processes can be the origin of a current sink or source (*Buzsáki, 2006*; *Pettersen et al., 2006*; *Einevoll et al., 2007*). For example, a current source may reflect an inhibitory synaptic current or an outflowing return current resulting from excitatory synaptic input elsewhere on the neuron. There is no simple way of knowing which it is from an extracellular recording alone (*Buzsáki, 2006*).

Another approach to uncovering the biophysical origins of current sinks and sources, and by extension the LFP, is to simulate them computationally (*Pettersen et al., 2008*; *Einevoll et al., 2013*). Following the classic work by Rall in the 1960s (*Rall, 1962*), a forward-modeling scheme in which extracellular potentials are calculated from neuron models with detailed morphologies using volume conduction theory under the line source approximation has been established (*Holt and Koch, 1999*). With this framework, we have achieved a good understanding of the biophysical origins of extracellular action potentials (*Koch, 1998*; *Holt and Koch, 1999*; *Pettersen and Einevoll, 2008*; *Hay et al., 2011*; *Lindén et al., 2010*). Expanding on this understanding, models composed of populations of unconnected neurons (e.g., *Pettersen et al., 2008*; *Lindén et al., 2011*; *Schomburg et al., 2012*; *Łęski et al., 2013*; *Sinha and Narayanan, 2015*; *Hagen et al., 2017*; *Ness et al., 2018*) and recurrent network models (e.g., *Traub et al., 2005*; *Vierling-Claassen et al., 2010*; *Reimann et al., 2013*; *Głąbska et al., 2014*; *Tomsett et al., 2015*; *Hagen et al., 2016*; *Hagen et al., 2018*; *Chatzikalymniou and Skinner, 2018*) have been used to study the neural processes underlying LFP.

While interesting insights about CSD and LFP were obtained from these computational approaches, establishing a direct relationship between the biological details of the circuit structure and the electrical signal like LFP remains a major unresolved challenge. One reason is that the amount and quality of data available for modeling the circuit architecture in detail have been limited. This situation improved substantially in recent years, and a broad range of data on the composition, connectivity, and physiology of cortical circuits have been integrated systematically (*Billeh et al., 2020*) in a biophysically detailed model of mouse primary visual cortex (area V1). In addition, significant improvements were achieved in experimental recordings of the LFP and the simultaneous spiking responses. In particular, the Neuropixels probes (*Jun et al., 2017*) record LFP and hundreds of units across the cortical depth in multiple areas, with 20 μm spacing between recording channels allowing for an unprecedented level of spatial detail. These developments provide unique opportunities to improve our understanding of circuit mechanisms that determine LFP patterns.

Here, we analyze spikes and LFP from the publicly available Allen Institute's *Visual Coding* survey recorded using Neuropixels probes (https://www.brain-map.org; *Siegle et al., 2021*) and reproduce these using the mouse V1 model developed by *Billeh et al., 2020*. The model is comprised of >50,000 biophysically detailed neuron models surrounded by an annulus of almost 180,000 generalized leaky-integrate-and-fire units. The neuron models belong to 17 different cell type classes: one inhibitory class (Htr3a) in layer 1, and four classes in each of the other layers (2/3, 4, 5, and 6) where one is excitatory and three are inhibitory (Pvalb, Sst, Htr3a) in each layer. The visual coding dataset consists of simultaneous recordings from six Neuropixels 1.0 probes across a range of cortical and subcortical structures in 58 mice while they are exposed to a range of visual stimuli (about 100,000 units and 2 billion spikes over 2 hr of recording).

In our analysis of this dataset, we identified a canonical CSD pattern that captures the evoked response in mouse V1 to a full-field flash. We then modified the biophysically detailed model of mouse V1 to reproduce both the canonical CSD pattern and laminar population firing rates in V1 simultaneously. We reproduce, in a quantitative manner, the shape and timing of the pattern of current sources and sinks that have been described in considerable detail by experimentalists (e.g., *Mitzdorf, 1987*; *Swadlow et al., 2002*; *Senzai et al., 2019*). This shows that adjustments to synaptic parameters such as weights and placement in addition to a circuit architecture that included feedback are sufficient to reproduce experimental findings on both single-cell measures such as spikes and population-level measures such as CSD. We use this model to explain, in a highly mechanistic manner, the biophysical origins of the various ionic current sinks and sources and their location across the various layers of visual cortex.

In the process of obtaining a model that could reproduce both spikes and CSD, we discovered that the model can be modified by adjusting the synaptic weights to reproduce the experimental firing rates with only minor effects on the simulated CSD, and, conversely, that the simulated CSD can be altered with only minor effects on the firing rates by adjusting synaptic placement. Furthermore, we found that comparing the simulated CSD to the experimental CSD revealed discrepancies between model and data that were not apparent from only comparing the firing rates. Additionally, it was not until feedback from higher cortical visual areas (HVAs) was added to the model that simulations reproduced both the experimentally recorded CSD and firing rates, as opposed to only the firing rates. This bio-realistic modeling approach sheds light on specific components of the V1 circuit that contribute to the generation of the major sinks and sources of the CSD in response to abrupt visual stimulation. Our findings demonstrate that utilizing the LFP and/or the CSD in modeling can aid model configuration and implementation by revealing discrepancies between models and experiments and provide additional constraints on model parameters beyond those offered by the spiking activity. The new model obtained here is freely accessible (https://doi.org/10.5061/dryad.k3j9kd5b8) to the community to facilitate further applications of biologically detailed modeling.

## Results

Spikes and LFP were recorded across multiple brain areas, with a focus on six cortical (V1, LM, AL, RL, AM, PM) and two thalamic (LGN, LP) visual areas, using Neuropixels probes in 58 mice (*Siegle et al., 2021*).

A schematic of the six probes used to perform the recordings in individual mice is shown in *Figure 1A*, and the spikes and LFP recorded in V1 of an exemplar mouse during presentation of a full-field bright flash stimulus are displayed in *Figure 1B, C*. The CSD can be estimated from the LFP (averaged over 75 trials) using the delta iCSD method to obtain a more localized measure of inflowing (sinks) and outflowing currents (sources) (*Pettersen et al., 2006*; *Einevoll et al., 2013*). The biophysically detailed model of mouse V1 used to simulate the neural activity and the recorded potential in response to the full-field flash stimulus is illustrated in *Figure 1E*. The model contains 230,924 neurons, of which 51,978 are biophysically detailed multicompartment neurons with somatic Hodgkin–Huxley conductances and passive dendrites, and 178,946 are leaky-integrate-and-fire (LIF) neurons. These neuron models are arranged in a cylinder with a radius 845 µm and a height 860 µm. The multicompartment neurons are placed in the 'core' with a radius of 400 µm, while the LIF neurons form an annulus surrounding this core. Cellular models belong to 17 different classes: one excitatory class and three inhibitory (Pvalb, Sst, Htr3a) in each of layers 2/3, 4, 5 and 6, and a single Htr3a inhibitory class in layer 1. The extracellular electric field in the model was recorded on an array of simulated

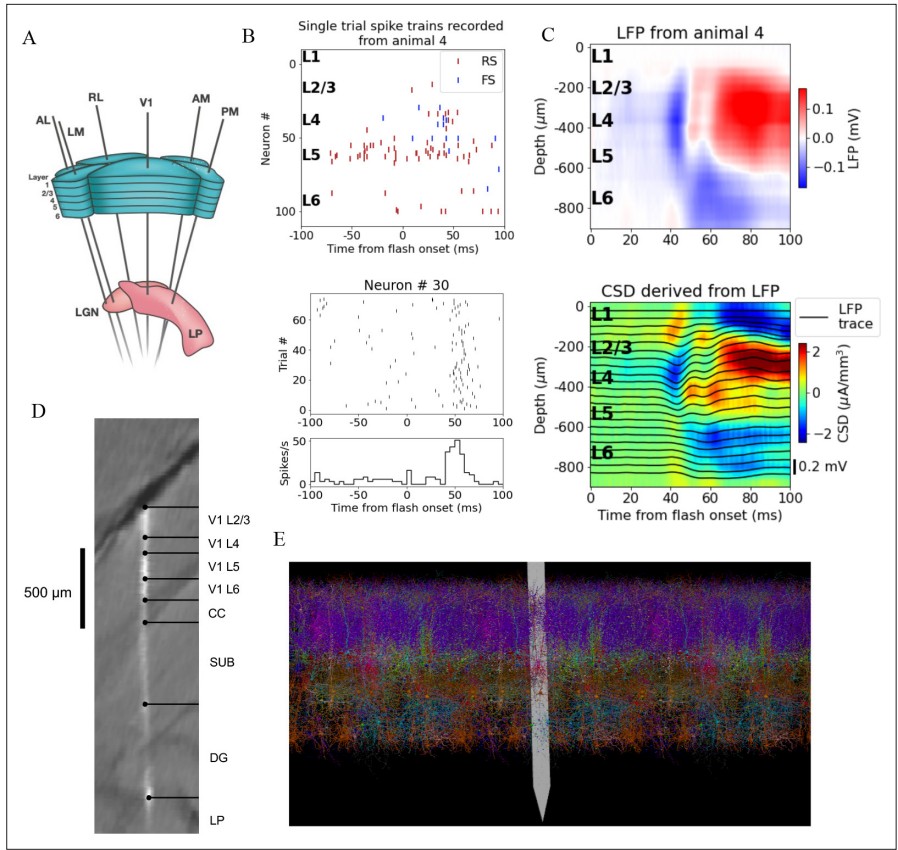

**Figure 1.** Illustration of experimental data and the biophysical model for mouse primary visual cortex (V1). (**A**) Schematic of the experimental setup, with six Neuropixels probes inserted into six cortical (V1, latero-medial [LM], rostro-lateral [RL], antero-lateral [AL], postero-medial [PM], AM) and two thalamic areas (LGN, LP). (**B**) Top: spikes from many simultaneously recorded neurons in V1 during a single trial. Bottom: spikes from a single neuron recorded across multiple trials. In both cases, the stimulus was a full-field bright flash (onset at time 0, offset at 250 ms). (**C**) Top: local field potential (LFP) across all layers of V1 in response to the full-field bright flash, averaged over 75 trials in a single animal. Bottom: current source density (CSD) computed from the LFP with the delta iCSD method. (**D**) Histology displaying trace of the Neuropixels probe across layers in V1, subiculum (SUB) and dentate gyrus (DG). (**E**) Visualization of the V1 model with the Neuropixels probe in situ. (Image made using VND.)

point electrodes (*Dai et al., 2020*) arranged in a straight line (*Figure 1D*) and separated by 20 μm, consistent with Neuropixels probes, shown in *Figure 1E* to scale with the model.

## Uncovering a canonical visually evoked CSD response

We first established a 'typical' experimentally recorded CSD pattern to be reproduced with the model. Though there is substantial inter-trial and inter-animal variability in the evoked CSD response, we find that most trials and animals have several salient features in common. In *Figure 2A*, the trial-averaged evoked CSDs from five individual mice are displayed. In the first four animals (#1–4), we observe an early transient sink arising in layer 4 (L4) ~40 ms after flash onset, followed by a sustained source starting ~60 ms, which covers L4 and parts of layers 2/3 (L2/3) and layer 5 (L5). We also observe a sustained sink covering layers 5 and 6 (L6) emerging around 50 ms, as well as a sustained sink covering layers 1 and 2/3 around 60 ms. An animal that does not fully exhibit what we term the 'canonical' pattern is shown in the rightmost plot (#5 in *Figure 2A*); it has an early L4 sink arising at 40 ms, but this sink is not followed by the sustained sinks and sources from 50 to 60 ms and onward observed in the other animals. The timing and location of sinks and sources are, overall, similar to those described earlier by *Givre et al., 1994*; *Schroeder et al., 1998*, *Niell and Stryker, 2008*, and *Senzai et al., 2019*.

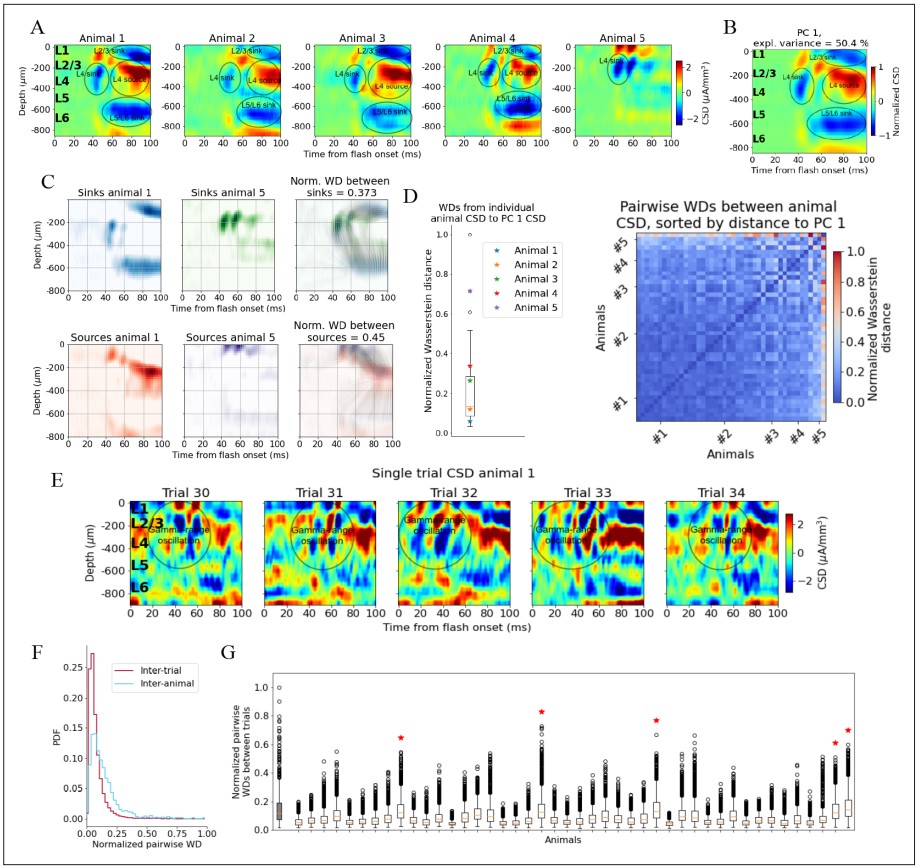

**Figure 2.** Variability in experimentally recorded current source density (CSD). (**A**) Evoked CSD response to a full-field flash averaged over 75 trials, from five animals in the dataset. (**B**) The first principal component (PC) computed from the CSD of all n = 44 animals, explaining 50.4% of the variance. (**C**) Illustration of movement of sinks and sources in the calculation of the Wasserstein distance (WD) between the CSD of two animals in the dataset. The gray lines in the rightmost panels display how the sinks or sources of one animal are moved to match the distribution of sinks or sources of the other animal. (**D**) Left: WDs from each animal to the PC 1 CSD. Right: pairwise WDs between all 44 animals sorted by their distance to the first PC. (**E**) CSD from five individual trials in example animal 1. (**F**) Distribution of pairwise distances between single-trial CSD (red) and pairwise distances between trial-averaged CSD of individual animals (blue). Both are normalized to the maximum pairwise distance between the trial-averaged CSD of individual animals. (**G**) Pairwise WDs between trials in each of 44 animals (white boxplots), normalized to maximal pairwise WDs between trial-averaged CSD of animals. Gray-colored boxplot shows the distribution of pairwise WDs between trial-averaged CSD of individual animals, and the red stars indicate the n = 5 animals for which the inter-trial variability was greater than the inter-animal variability (assessed with Kolmogorov–Smirnov [KS] tests, p < 0.001 in all cases, see *Figure 2—figure supplement 3*).

The online version of this article includes the following figure supplement(s) for figure 2:

**Figure supplement 1.** Trial-averaged current source density (CSD) during presentation of full-field flashes for all 44 animals in this study.

**Figure supplement 2.** Principal component analysis (PCA) on histology-aligned current source density (CSD).

**Figure supplement 3.** Comparing inter-trial and inter-animal pairwise Wasserstein distances (WDs).

To identify the robust features across animals in this dataset, we performed principal component analysis (PCA) on the trial-averaged evoked CSD from all animals. Out of the 58 animals in the dataset, 5 did not have readable recordings of LFP in V1 during the presentation of the full-field flash stimuli, and the exact probe locations in V1 could not be recovered for 9 other animals due to fading of fluorescent dye or artifacts in the optical projection tomography (OPT) volume (see 'Materials and methods'). The remaining 44 (out of the 58) animals in the dataset were retained for the CSD analysis. The trial-averaged CSD plots of all these 44 animals are displayed in *Figure 2—figure supplement 1*. The first principal component (PC 1) (*Figure 2B*) constitutes a sum of weighted contributions of the

CSD patterns from all 44 animals and explains half (50.4%) of the variance. The salient features typically observed in individual animals are also prominent in the PC 1 CSD pattern (*Figure 2B*), that is, the canonical pattern. In *Figure 2—figure supplement 2*, the first 10 principal components cumulatively explaining 90% of the variance are plotted.

## Quantifying CSD pattern similarity

We use the Wasserstein, or Earth Mover's, distance (WD) to quantify the differences in CSD patterns (see 'Materials and methods'), which can then be used to assess how well the simulated CSD matches the CSD typically observed in experiments. The WD reflects the cost of transforming one distribution into another by moving its 'distribution mass' around (*Rubner et al., 1998*; *Arjovsky et al., 2017*). An often-used analogy refers to the two distributions as two piles of dirt, where the WD tells us the minimal amount of work that must be done to move the mass of one pile around until its distribution matches the other pile (*Rubner et al., 1998*). In the context of CSD patterns, the WD reflects the cost of transforming the distribution of sinks and sources in one CSD pattern into the distribution of sinks and sources in another pattern, with larger WD indicating greater dissimilarity between CSD patterns. The WDs are computed between the sinks of two CSD patterns and between the sources of two CSD patterns independently, and then summed to form a total WD between the CSD patterns (*Figure 2C*). The sum of all sinks and the sum of all sources in each CSD pattern are normalized to −1 and +1, respectively, so the WD only reflects differences in patterns, and not differences in the overall amplitude. The WD scales linearly with shifts in space and time.

When computing the WDs between the evoked CSD patterns of individual animals and the canonical pattern, we find that the animals with CSD patterns that, by visual inspection, resemble the canonical pattern (*Figure 2A*, animals 1–4), are indeed among animals with smaller WD, while the animal with the more distinct CSD pattern (*Figure 2A*, animal 5) is an outlier (*Figure 2D*).

The onset of the evoked response is less conspicuous in the single-trial CSD due to pronounced, ongoing sinks and sources, but there is still a visible increase in magnitude from 40 to 50 ms onward (*Figure 2E*), compatible with the latency of spiking responses to full-field flashes in V1 (*Siegle et al., 2021*). An oscillation of sinks and sources with a periodicity of ~20 ms, that is, in the gamma range is apparent in the region stretching from L2/3 to the top of L5, which appears to be either partially interrupted or drowned out by more sustained sinks and sources emerging at about 60 ms. At least some of this gamma-range activity derives from the visual flash that covers the entire visual field and that drives retinal neurons and postsynaptic targets in the lateral geniculate nucleus (LGN) in an oscillatory manner (see the pronounced gamma-range oscillation in the LGN firing rate in *Figure 3D*).

The inter-trial variability is roughly comparable to the inter-animal variability of the trial-averaged responses. By computing the pairwise Wasserstein distances between single trial CSDs within each animal and comparing it to the pairwise WD between the trial-averaged CSD of each animal, we find that inter-trial variability in CSD is significantly lower than the inter-animal variability in trial-averaged CSD (Kolmogorov–Smirnov distance = 0.33; p<0.001) (*Figure 2F*).

The majority of animals (39 out of 44) have a WD to the first principal component, PC 1, of the CSD that is less than half of the greatest WD between the CSD of individual animals and the PC 1 CSD (*Figure 2D*); the pairwise WDs between animals are also less than half of the maximum pairwise WD for most animals (921 out of the total 946 pairwise WDs; *Figure 2E*). This supports the view that most animals exhibit the canonical CSD pattern captured by the PC 1 CSD (*Figure 2B*). The total inter-trial variability is smaller than the inter-animal variability, both estimated by pairwise WDs (*Figure 2F and G*), though there are n = 5 animals for which the inter-trial WDs are larger than the inter-animal WDs (*Figure 2G*, marked by red stars; determined with KS tests on the distribution of pairwise WDs between animals and pairwise WDs between trials in each animal; see *Figure 2—figure supplement 3*).

## Quantifying firing rate variability

For the spike analysis, we distinguish between fast-spiking (FS; putative Pvalb inhibitory) neurons and regular-spiking (RS; putative excitatory and non-Pvalb inhibitory) neurons (see 'Materials and methods' and *Figure 3—figure supplement 1*). All FS-neurons are grouped together into one population across all layers, while the RS-neurons are divided into separate populations for each layer (*Figure 3A*). The FS-neurons are merged across layers because we set a criterion of at least 10 recorded neurons in any

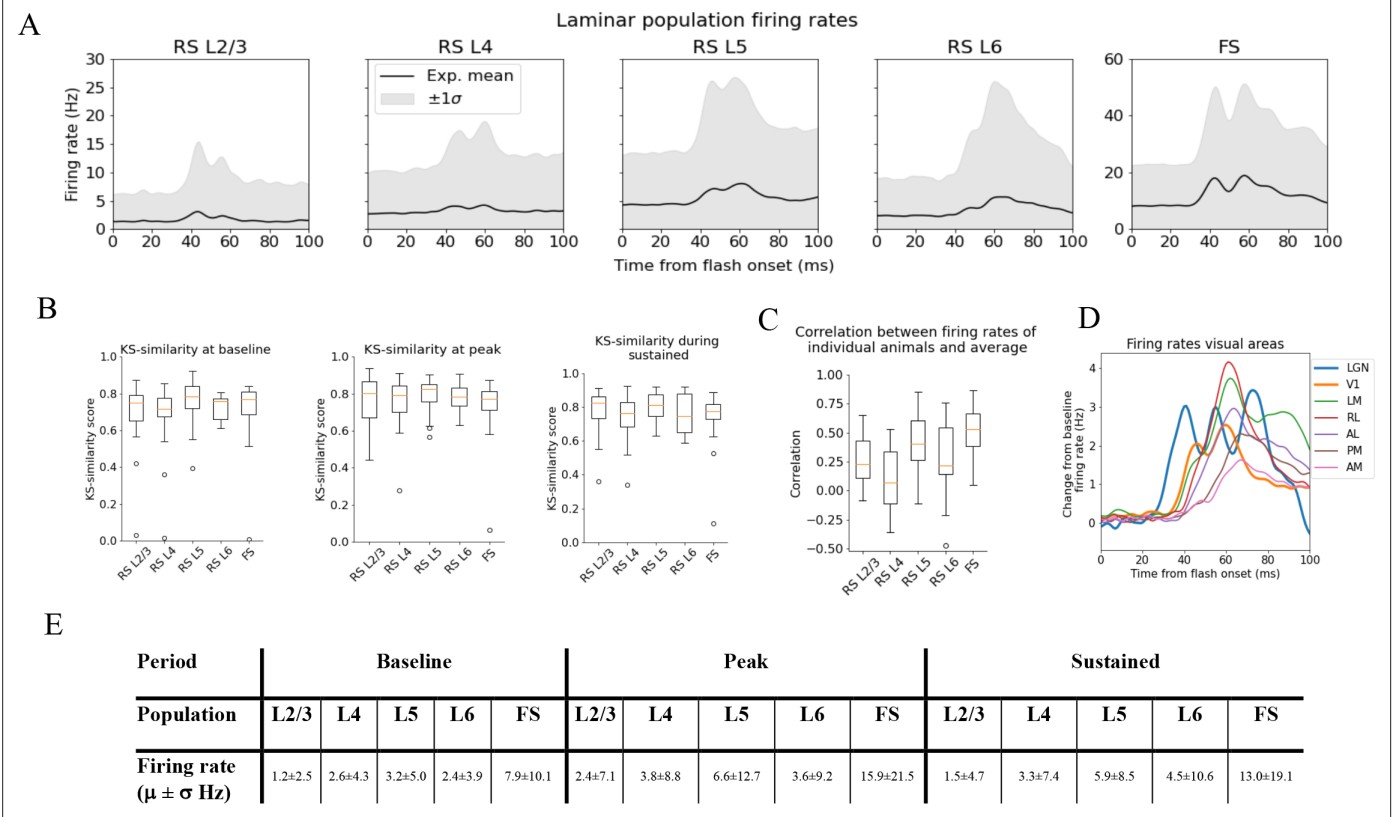

**Figure 3.** Variability in experimentally recorded spikes. (**A**) Trial-averaged laminar population firing rates of regular-spiking (RS) cells, differentiated by layer, and fast-spiking (FS) cells across all layers in response to full-field flash. Black line: average across all animals. Gray shaded area: ±1 standard deviation. (**B**) Kolmogorov–Smirnov (KS) similarities (see 'Materials and methods') between the trial-averaged firing rates of each individual animal and the average firing rate over cells from all animals (black line in **A**) at baseline (the interval of 250 ms before flash onset), peak evoked response (from 35 to 60 ms after flash onset), and during the sustained period (from 60 to 100 ms). (**C**) Correlations between trial-averaged firing rates of individual mice and all mice (0–100 ms after flash onset). (**D**) Baseline-subtracted evoked firing rates for excitatory cells in seven visual areas (average over trials, neurons, and mice). Note the strong, stimulus-triggered gamma-range oscillations in the firing of lateral geniculate nucleus (LGN) neurons (blue). (**E**) Mean (μ) ± standard deviation (σ) of population firing rates during baseline, peak evoked response, and the sustained period. Averaged across trials, neurons and time windows defined above.

The online version of this article includes the following figure supplement(s) for figure 3:

**Figure supplement 1.** Classifying cell types in experimental data.

**Figure supplement 2.** Number of cells in each population in experimental data.

one layer when comparing the population firing rate in individual animals to the average population firing rate in all animals, and only *one* animal had 10 FS-neurons or more in any layer (*Figure 3—figure supplement 2*). This criterion was set to have a more reliable estimate of the population firing rates in individual animals.

We use the KS similarity (defined as one minus the KS distance, see 'Materials and methods') and correlation to quantify the variability in firing rates. We use the experimental variability as a reference to assess whether the model reproduces firing rates typically observed in experiments. The KS similarity gives the similarity between the distributions of average firing rates across neurons in two populations in selected time windows, with KS similarity = 1 implying identity. As such, KS similarity provides a metric to compare the magnitudes of firing rates in certain time periods. We defined the 'baseline' window as the period over 250 ms before the flash onset, the 'initial peak' window as 35–60 ms after flash onset, and the 'sustained' window as 60–100 ms after flash onset. The KS similarity score during baseline is denoted 'KSS$_b$,' during the 'initial peak' 'KSS$_p$,' and 'sustained' 'KSS$_s$.' The correlation, on the other hand, is computed between two population firing rates throughout the 100 ms window. The correlation thus gives us a measure of the similarity in the temporal profile of firing rates in this interval, independent of magnitudes. We establish the experimental variability in KS similarities

and correlation by computing these metrics between the population firing rates of each individual animal and the average population firing rates of all other animals (averaged over trials for both the individual animals and the average over all other animals) (*Figure 3B and C*).

The population firing rates for FS neurons are more than twice as high as RS cells during baseline, peak, and sustained. Among the RS populations, the firing rate in L5 is the highest in all periods, followed by L4 and L6, while L2/3 has the lowest firing rates (*Figure 3E*).

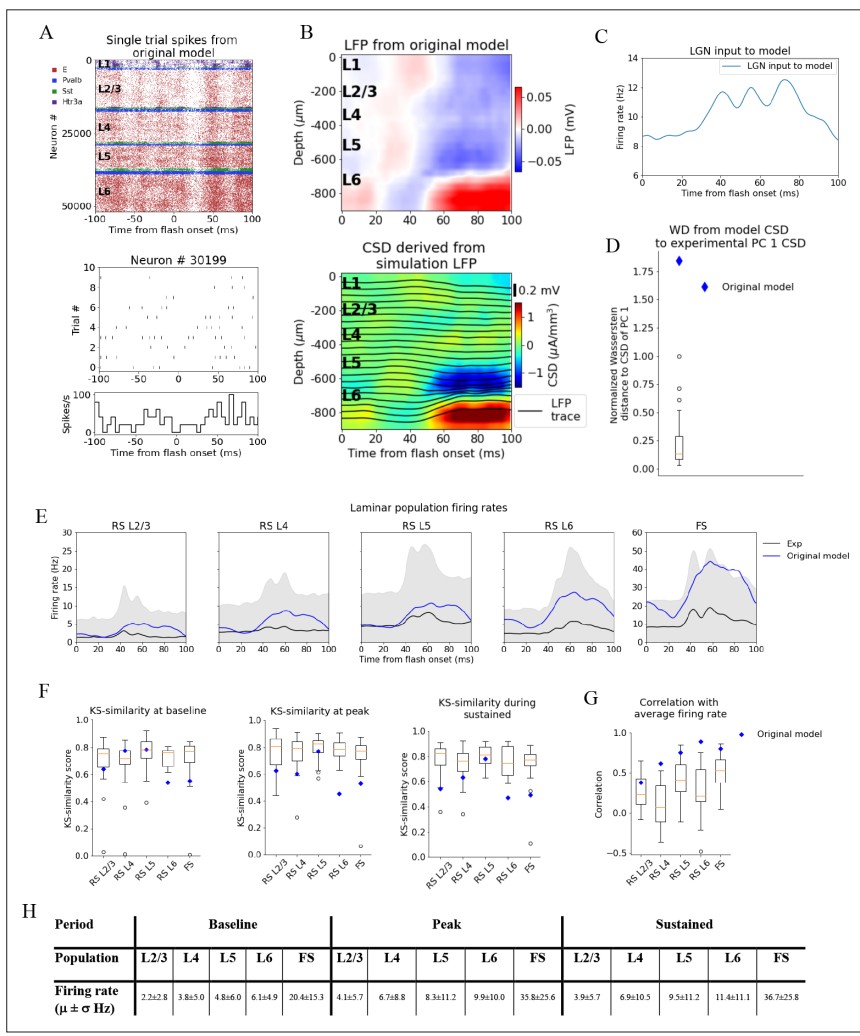

| Period | Baseline | | | | | Peak | | | | | Sustained | | | | |
|---|---|---|---|---|---|---|---|---|---|---|---|---|---|---|---|
| Population | L2/3 | L4 | L5 | L6 | FS | L2/3 | L4 | L5 | L6 | FS | L2/3 | L4 | L5 | L6 | FS |
| Firing rate (μ ± σ Hz) | 2.2±2.8 | 3.8±5.0 | 4.8±6.0 | 6.1±4.9 | 20.4±15.3 | 4.1±5.7 | 6.7±8.8 | 8.3±11.2 | 9.9±10.0 | 35.8±25.6 | 3.9±5.7 | 6.9±10.5 | 9.5±11.2 | 11.4±11.1 | 36.7±25.8 |

**Figure 4.** Local field potential (LFP), current source density (CSD), and spikes from simulations with the original model. (**A**) Top: raster plot of all ~50,000 cells in the model's 400 μm radius 'core' region spanning all layers, in a simulation of a single trial with the flash stimulus. Bottom: raster plot and histogram of spikes from 10 trials for an example cell. (**B**) Top: simulated LFP averaged over 10 trials of flash stimulus. Bottom: CSD calculated from the LFP via the delta iCSD method. (**C**) Firing rate of experimentally recorded lateral geniculate nucleus (LGN) spike trains used as input to the model. (**D**) Wasserstein distance between CSD from the original model (blue diamond) and PC 1 CSD from experiments together with the Wasserstein distances from experimental CSD in every animal to PC 1 CSD (boxplot), normalized to maximal distance for animals. (**E**) Experimentally recorded firing rates (black) and simulated firing rates (blue). (**F**) Kolmogorov–Smirnov (KS) similarity between firing rates in original model (blue diamond) or individual animals (boxplots) and firing rates in experiments at baseline, peak evoked response, and during the sustained period (defined in *Figure 3*). (**G**) Correlation between firing rates of model (blue diamond) or individual animals in experiments (boxplots) and average population firing rates in experiments (0–100 ms). (**H**) Mean (μ) ± standard deviation (σ) of model firing rates during baseline, peak evoked response, and the sustained period. Averaged across trials, neurons and time windows defined above.

The online version of this article includes the following figure supplement(s) for figure 4:

**Figure supplement 1.** Effect of reducing recurrent inhibition.

## Discrepancy between the original model and experimental observations

We simulated the response to a full-field flash stimulus with the biophysical network model of mouse primary visual cortex as presented in *Billeh et al., 2020*. As input to the model, we used experimentally recorded LGN spike trains (*Figure 4C*; see 'Materials and methods'). A Poisson source, firing at a constant rate of 1 kHz, provides additional synaptic input to all cells, representing the influence from the rest of the brain ('background' input). The thalamocortical input consists of spike trains from 17,400 LGN units (*Arkhipov et al., 2018*; *Billeh et al., 2020*). The public Neuropixels data contain recordings from 1263 regular-spiking LGN neurons across 32 mice during 75 trials of full-field bright flash presentations, resulting in 94,725 spike trains. To construct the input for each of our 10 simulation trials, we randomly sampled 10 unique subsets of spike trains from this pool until all 17,400 units had been assigned a spike train in each trial.

*Figure 4A, B* displays the resulting spiking pattern across all layers with its associated LFP. The inferred CSD exhibits a strong sink in the L5 and L6 region, matched by a strong source below it, both starting at ~50 ms after flash onset (*Figure 4B*, bottom). However, the early L4 sink, the later sustained L4 source, and the sustained L2/3 sink typically observed in the experimental CSD (*Figure 2A, B*) are either absent or too weak compared to the sink and source in L5 and L6. The WD from the simulated CSD to the experimental PC 1 CSD is greater than the WD between the CSD of the farthest outlier animal and the PC 1 CSD (WD = 1.84, normalized to the largest WD between CSD of individual animals and PC 1 CSD). Thus, using experimental variability as a reference, the CSD from this simulation is an outlier (*Figure 4C*).

The population firing rates of the model, the KS similarities and correlation between the model and the data, are plotted together with the data in *Figure 4D–F*. The magnitudes of the model firing rates are higher than the experimental firing rates in all populations and time windows (*Figure 4H*). However, the KS similarities between the model firing rates and the experimental firing rates are still within the minimum to maximum range of the boxplots for the RS L2/3, RS L4, and RS L5 cells in all time windows (*Figure 4F*), and during baseline for the FS cells. For RS L6 neurons, the KS similarities were among the outliers of the experiments in all time windows, while for FS neurons they were among the outliers during the peak and sustained windows (RS L2/3: $KSS_b = 0.62$, $KSS_p = 0.63$, and $KSS_s = 0.54$; RS L4: $KSS_b = 0.77$, $KSS_p = 0.60$, and $KSS_s = 0.63$; RS L5 $KSS_b = 0.77$, $KSS_p = 0.77$, and $KSS_s = 0.78$; RS L6: $KSS_b = 0.54$, $KSS_p = 0.45$, and $KSS_s = 0.47$; FS: $KSS_b = 0.54$, $KSS_p = 0.53$, and $KSS_s = 0.49$). The temporal profile of the model firing rates is above the minimum of the boxplots for all populations (RS L2/3: $r = 0.38^{***}$, RS L4: $r = 0.62^{***}$, RS L5: $r = 0.75^{***}$, RS L6: $r = 0.90^{***}$, FS: $r = 0.80^{***}$; $^{***}p<0.001$). For RS L4 and RS L6, the model is in fact outside the experimental distribution, but it is an outlier in a positive sense; the model firing rates for these populations are more similar to the experimental average than the corresponding population firing rates in individual animals.

The original model studied in *Figure 4* produced firing rates and orientation and direction tuning consistent with recordings in vivo (*Billeh et al., 2020*) with some shortcomings, such as relatively slow responses of V1 to the onset of visual stimuli (*Arkhipov et al., 2018*; *Billeh et al., 2020*). Here, we see even more inconsistencies reflected clearly in the CSD pattern. This demonstrates the importance of multi-modal characterization of such biologically detailed models. To investigate the properties of the cortical circuit that sculpt the CSD, we manipulated the model and observed how both the CSD and firing rate responses were improved to match the experimental data.

## Adjusting the model to fit experimental firing rates

Due to the discrepancy between the magnitudes of the model firing rates and the experimental firing rates, especially with respect to the outliers of the modeled RS L6 and FS neurons, we selectively adjusted the recurrent synaptic weights. We left the synaptic weights between LGN and the V1 model unchanged since they were well constrained by data (*Billeh et al., 2020*).

We first reduced the synaptic weights from all excitatory populations to the FS PV-neurons by 30% to bring their firing rates closer to the average firing rate in this population in the experiments. This resulted in increased firing rates in all other (RS) populations due to the reduced activity of the inhibitory Pvalb-neurons (*Figure 4—figure supplement 1*). Therefore, we further applied reductions in the synaptic weights from all excitatory neurons to RS neurons and increases in the synaptic weights from inhibitory neurons to the RS neurons to bring their firing rates closer to the experimental average

firing rates. We multiplied the recurrent synaptic weights with factors in the [0.2, 2.5] range until we arrived at a set of weights where none of the model firing rates were among the experimental outliers in any time window ($KSS_b$ = 0.73, $KSS_p$ = 0.77, and $KSS_s$ = 0.70; average across RS populations and the FS population) and temporal profiles (RS L2/3: $r$ = 0.49***, RS L4: $r$ = 0.63***, RS L5: $r$ = 0.71***, RS L6: $r$ = 0.87***, FS: $r$ = 0.86***; *** p<0.001) (*Figure 5A–C*).

The resulting pattern (but not the magnitude) of the CSD, however, was largely unchanged (*Figure 5D*) compared to the original CSD (*Figure 4B*). The overall magnitude was reduced, and there were some traces of a sink arising at 40 ms after flash onset, and a L2/3 (and L1) sink after 60 ms, but they were substantially weaker relative to the L5/L6 dipole than they were in the experiments. Furthermore, the large and sustained L4 source after 60 ms was still either absent or too weak to be visible. The WD between the CSD from this version of the model and the experimental PC 1 CSD remained among the outliers of the animals (*Figure 5E*) (normalized WD = 1.26).

## Two-way dissociation between spikes and CSD

Simulations demonstrate that the LFP, and the associated CSD, can be significantly altered by changes to synaptic placement (*Einevoll et al., 2007*; *Pettersen and Einevoll, 2008*; *Lindén et al., 2010*; *Lindén et al., 2011*; *Łęski et al., 2013*; *Hagen et al., 2017*; *Ness et al., 2018*). As observed in *Figure 5A–E* and *Figure 5—figure supplement 1*, adjustments to synaptic weights can modify the population firing rates substantially, yet without substantially changing the pattern of the CSD, that is, the placement and timing of sinks and sources. The inverse can also occur; that is, the CSD pattern can be altered extensively with only minor effects on firing rates (*Figure 5F and G*, *Figure 5—figure supplements 2–4*).

In the model's original network configuration, L4 excitatory neurons received geniculate input from synapses placed within 150 µm from the soma on both basal and apical dendrites, and excitatory, recurrent input from other V1 neurons within 200 µm from the soma on both basal and apical dendrites. We tested the effects of synaptic location by placing all synapses from both LGN and excitatory neurons onto the basal dendrites of L4 excitatory neurons (within the same ranges as in the original configuration). This increased the contribution from the L4 excitatory neurons to the total CSD (*Figure 5F*, middle row, leftmost plot) by a factor of ~2 and led to a dipole pattern with a single sink at the bottom and a single source at the top, as opposed to having two pairs of sinks and sources like in the case of the original synaptic placement (*Figure 5F*, top row; leftmost plot). The firing rate of the L4 excitatory cells, however, remained essentially unchanged by this modification (*Figure 5G*). On the other hand, placing all synapses from LGN and excitatory neurons onto the apical dendrites of L4 excitatory neurons resulted in even greater CSD magnitude from this population (*Figure 5F*, bottom row; leftmost plot), while the magnitude of its firing rates were reduced (*Figure 5G*). In this case, the pattern displayed a sink in the middle with a source above and below it. We also find that the somatic Hodgkin–Huxley channels significantly shapes the CSD pattern (*Figure 5H*), generating a source in the L4 region where the somata are localized.

We quantified the changes in CSD of the full model resulting from changes in recurrent synaptic weights in *Figure 5—figure supplement 1* and the changes in firing rates resulting from the changes in synaptic placement onto L4 excitatory cells in *Figure 5—figure supplement 2*. Our analyses corroborate that the changes in CSD after changing only synaptic weights is minor, and that the changes in firing rate after changing only synaptic placement is minor. More examples of the disparate effects on CSD and spikes from adjusting placement of excitatory synapses on excitatory cells in L2/3 and L5 are displayed in *Figure 5—figure supplement 3* and *Figure 5—figure supplement 4*, respectively.

These results indicate a two-way dissociation that can occur between CSD and firing rates of excitatory neurons. The firing rates can be changed without substantially changing the CSD by modifying the strength of synapses, while the CSD can be changed without substantially changing the firing rates by modifying synaptic location. This suggests that utilizing the CSD in the optimization of the model can provide constraints on the circuit architecture that could not be obtained from spikes alone.

## Effects of feedback from higher visual areas to the model

*Hartmann et al., 2019* found that feedback from higher visual areas (HVAs) can exert a powerful influence on the magnitude of the evoked LFP response recorded in V1 of macaque monkeys, particularly in the period 80–100 ms after stimulus onset. The sustained L2/3 sink and L4 source we observe in the

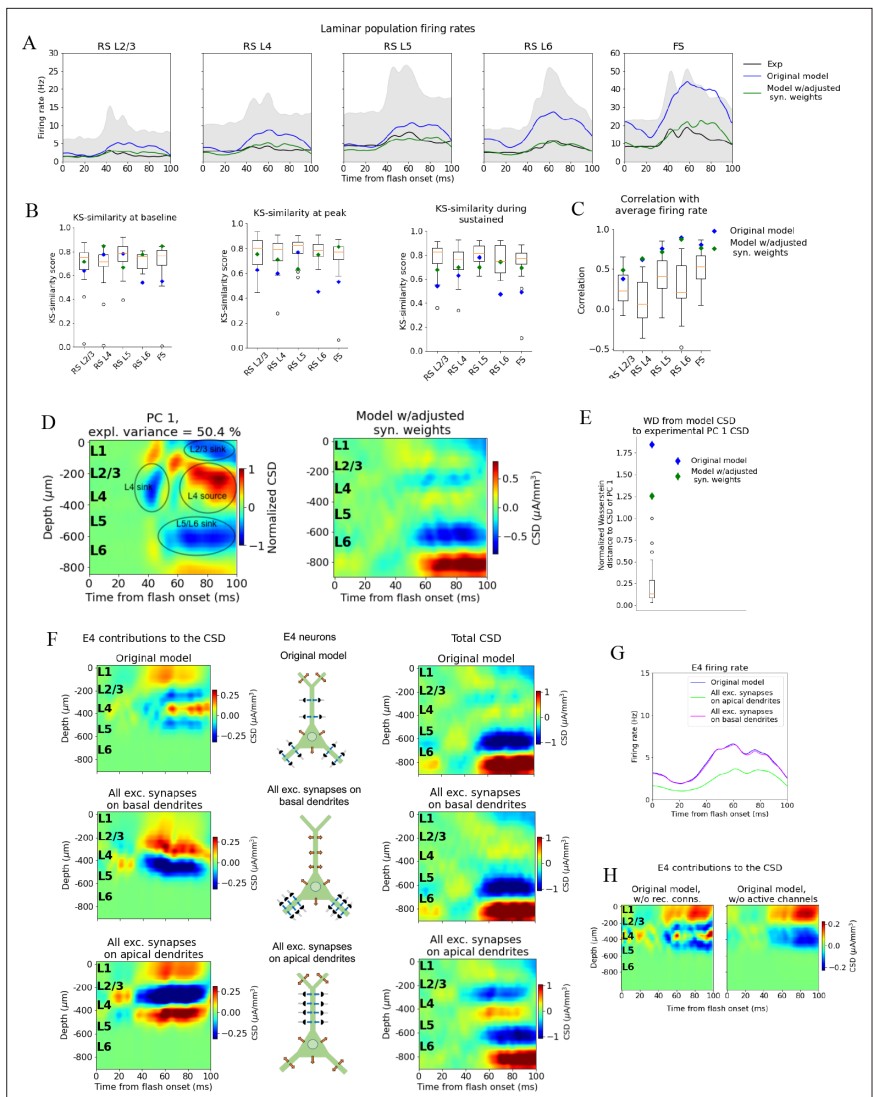

**Figure 5.** Adjusting the model to fit spikes or current source density (CSD). (**A**) Average experimentally (black) and simulated firing rates of experiments in the model with adjusted recurrent synaptic weights (green) and original model (blue). Synaptic adjustments included scaling the weights from all excitatory populations to the PV cells down by 30% to reduce the firing rates in these fast-spiking populations, reducing the synaptic weights from excitatory populations to all others and increasing synaptic weights from all PV cells to all other populations to compensate for the reduced inhibition. (**B**) Kolmogorov–Smirnov (KS) similarity between firing rates of model versions (markers) or individual animals in experiments (boxplots) and firing rates of experiments at baseline, peak evoked response, and during the sustained (defined in *Figure 3*). (**C**) Correlation between simulated firing rates or individual animals (boxplots) and measured firing rates (0–100 ms). (**D**) Left: PC 1 current source density (CSD) from experiments (see *Figure 2*). Right: CSD resulting from simulation on model with adjusted recurrent synaptic weights. (**E**) Wasserstein distance between CSD from model versions and PC 1 CSD from experiments together with Wasserstein distances from CSD in animals to PC 1 CSD (boxplot). (**F**) Effect of different patterns of placing excitatory synapses onto layer 4 excitatory cells on this population's contribution to the simulated CSD (left) and to the total simulated CSD (right). These synaptic placement schemes with accompanying inflowing (blue arrows) and outflowing (orange arrows) currents are illustrated in the middle. (**G**) Effect of synaptic placement on the simulated population firing rate. (**H**) Contribution of L4 excitatory cells to the simulated CSD in the model where all recurrent connections have been cut (left) and when all active channels have been removed from all cells in the model (right).

The online version of this article includes the following figure supplement(s) for figure 5:

**Figure supplement 1.** Quantifying change in simulated current source density (CSD) with adjustments to synaptic weights.

*Figure 5 continued on next page*

*Figure 5 continued*

**Figure supplement 2.** Quantifying change in spiking of L4 excitatory cells after adjusting synaptic placement.

**Figure supplement 3.** Effects of manipulating synaptic placement onto L2/3 excitatory cells on population current source density (CSD) and spiking.

**Figure supplement 4.** Effects of manipulating synaptic placement onto L5 excitatory cells on population current source density (CSD) and spiking.

experimental CSD emerge at 60 ms (*Figure 2A and B*), which roughly coincides with the peak firing rates in the latero-medial (LM), rostro-lateral (RL), antero-lateral (AL), and postero-medial (PM) cortical areas (*Figure 3C*). Furthermore, anatomical data indicate that synapses from HVAs terminate on L1 and L2/3 apical dendrites of pyramidal cells (whose cell bodies reside in L2/3 or L5) (*Glickfeld and Olsen, 2017*; *Marques et al., 2018*; *Hartmann et al., 2019*; *Keller et al., 2020*; *Shen et al., 2020*). Together, these observations suggest that the sustained L2/3 sink and L4 source might, in part, be induced by feedback from HVAs, where the sink is generated from the input to the apical tufts in L1 and L2/3, and the source may be the return currents of this input.

Of these HVAs, the feedback from LM to V1 is best characterized (*Marques et al., 2018*; *Keller et al., 2020*; *Shen et al., 2020*) and has the highest connection density to V1 (*Harris et al., 2019*). Based on these considerations, we decided to test the hypothesis that the large sinks and sources in the upper layers were caused, at least in part, by feedback from LM. In addition to the earlier feed-forward LGN input and the background input representing the influence of the rest of the brain, we introduced a feedback input constructed from experimentally recorded spike trains in LM. In total, the public Neuropixels dataset has 2075 neurons recorded in LM (simultaneously with the recordings in LGN, V1, and other visual areas) from 42 animals during presentations of the full-field flash stimulus. 1823 of the 2075 neurons were classified as RS, and spike trains from these were used to generate the feedback input to the model (*Figure 6A*).

The synapses from this LM source were placed on the apical dendrites of L2/3 excitatory neurons (within 150 μm from the soma), on the apical tufts (>300 μm from the soma) and the basal dendrites (within 150 μm from the soma) of L5 excitatory neurons, and on the somata and basal dendrites of L2/3, and L5 inhibitory (Pvalb and Sst) neurons (at any distance from the soma). The input onto L2/3 excitatory neurons did generate a sink in L1 and L2/3 and a source below in L4 (*Figure 6B–E*).

The synaptic weights from LM to the populations targeted by the feedback were initialized at high values (see 'Materials and methods'), and then adjusted (decreased) by multiplying them with factors in the range [0.05, 0.5] (see 'Materials and methods'). The weights from the background to the feedback-targeted populations were also multiplied by factors in the range [0.2, 0.5], and the weights of connections from Pvalb neurons to L2/3 excitatory and L5 excitatory neurons were multiplied by factors in the range [0.8, 1.2]. This weight scaling was done until the population firing rates were within the experimental variability. Additionally, the synapses from excitatory populations onto L6 excitatory cells were restricted to be within 150 μm from the soma to reduce the magnitude of the L5/L6 dipole (*Figure 6—figure supplement 1*; see 'Materials and methods'). Greater separation between a sink and a source in a dipole moment increases its magnitude, so restricting the range within which synapses can be placed should diminish the L5/L6 dipole's dominance.

When the model received this feedback input together with the LGN input, the resulting CSD pattern reproduced the main features observed in the experiments (*Figure 6C*). The WD between the model CSD and the experimental PC 1 CSD was also no longer an outlier (Normalized WD = 0.41; *Figure 6D*), and the population firing rates remained within the minimum and maximum value of the experimental boxplots for the firing rates in all windows and all populations, both with respect to magnitudes ($KSS_b = 0.77$, $KSS_p = 0.70$, and $KSS_s = 0.68$; average across all populations) and temporal profiles (RS L2/3: $r = 0.36$***, RS L4: $r = 0.64$***, RS L5: $r = 0.69$***, RS L6: $r = 0.87$***, FS: $r = 0.77$***, ***$p<0.001$) (*Figure 6F and G*). Thus, when average responses to the full-field flash are considered, this final adjusted model exhibits both the CSD and firing rate patterns that are consistent with the experimental observations and are well within animal variability (*Figure 6F–H*).

Furthermore, we investigated whether the model could reproduce the stereotypical features of the single-unit firing response observed in experiments. To this end, we computed the moments of the distribution of peak firing rates as well as the distribution of latencies to the peak across cells both for individual animals and for the model versions (*Figure 6—figure supplements 2 and 3*). The original

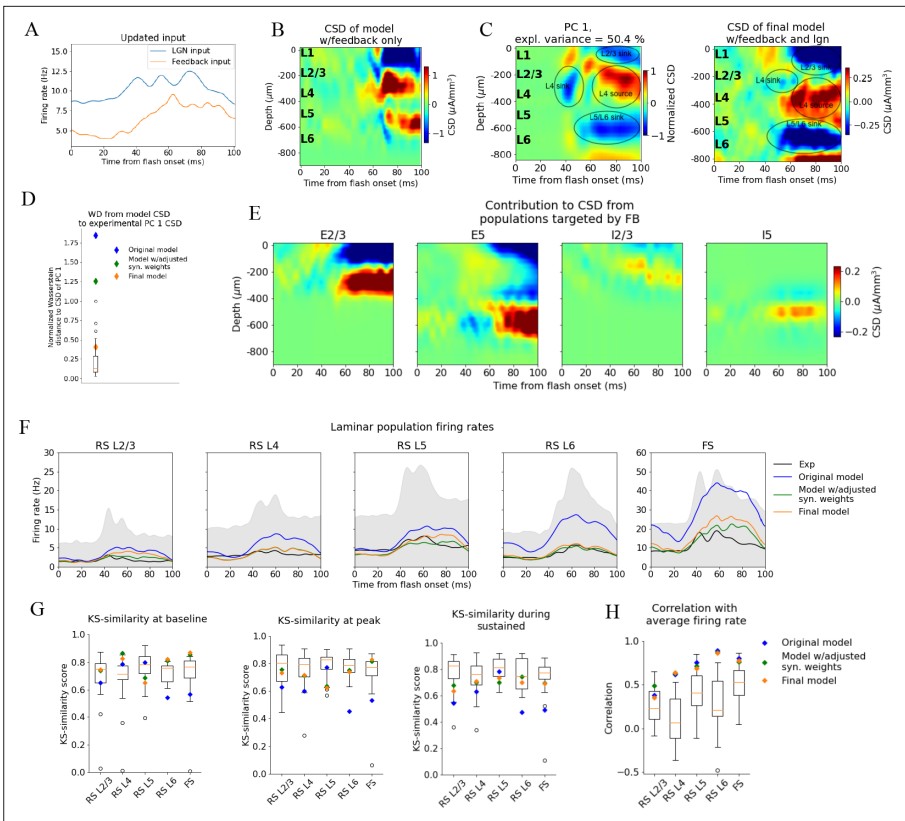

**Figure 6.** Introducing feedback from latero-medial (LM) to V1 in the model. (**A**) Firing rate of the experimentally recorded lateral geniculate nucleus (LGN) and LM units used as input to the model. (**B**) Total current source density (CSD) resulting from simulation with input only from the LM. (**C**) Left: PC 1 CSD from experiments (see *Figure 2*). Right: total CSD from simulation with both LGN input and LM input. (**D**) Wasserstein distance between CSD from model versions and PC 1 CSD from experiments together with Wasserstein distances from CSD in animals to PC 1 CSD (boxplot). (**E**) Population contributions from populations that receive input from LM. (**F**) Average population firing rates of experiments (black line) and model versions. (**G**) Kolmogorov–Smirnov (KS) similarity between simulated firing rates or individual animals (boxplots) and recorded firing rates at baseline, peak evoked response, and the sustained period (defined in *Figure 3*). (**H**) Correlation between simulated and experimentally recorded firing rates (0–100 ms).

The online version of this article includes the following figure supplement(s) for figure 6:

**Figure supplement 1.** Effect of adjusting synaptic placement onto L6 excitatory cells.

**Figure supplement 2.** Moments of distributions of peak firing rate in model versions and experiments for different populations.

**Figure supplement 3.** Moments of distributions of latency to peak of firing rates in model versions and experiments in different populations.

**Figure supplement 4.** Relative change in peak firing rates between neighboring populations.

**Figure supplement 5.** Moments of distributions of greatest curvature in firing rate across cells.

**Figure supplement 6.** Orientation and direction selectivity in final model.

**Figure supplement 7.** Current source density (CSD) analysis after aligning experimental CSD plots to landmarks rather than histology.

**Figure supplement 8.** Priincipal component analysis (PCA) on landmark aligned current source density (CSD).

**Figure supplement 9.** Effect of using plain average of trial-averaged current source density (CSD) from all animals instead of first principal component as target.

model was an outlier for the first moment of the peak firing rate distributions of RS L2/3, L4, L6, and FS populations, while the intermediate model was within the experimental variability for all populations, and the final model was within the experimental variability for all populations except RS L2/3, where it was just outside the maximum value of the experimental boxplot (*Figure 6—figure supplement 2A*). All three model versions were outside the experimental variability of the third moment (skewness) of the peak firing rate distributions for the RS L2/3 population, and the original model was just outside for the RS L5 and FS populations as well (*Figure 6—figure supplement 2C*). Otherwise, all model versions were within the experimental variability for all moments that we considered and for all populations.

With respect to the distribution of latencies to the peak firing rates, all model versions were within the experimental variability for all populations and all moments except the fourth moment (kurtosis) of the FS population, where all model versions were outside the experimental variability. We also computed the relative change in firing rates between neighboring populations (*Figure 6—figure supplement 4*) and the distributions of greatest curvature of the firing rate across cells (*Figure 6—figure supplement 5*), and found that all model versions were within the experimental variability on both of these metrics too. Thus, as with the analysis of population firing rates, the original model is an outlier when compared to the experiments on the features of unit firing, while the intermediate and final model versions reproduce most features observed in the experiments.

To check if the model continued to exhibit appropriate orientation and direction tuning after the adjustments made, we ran a simulation with the final model configuration and the same drifting grating stimulus that was utilized in *Billeh et al., 2020*. We found that the model still displayed firing rates at preferred directions and direction and orientation selectivity indices comparable to those observed experimentally (*Figure 6—figure supplement 6*).

## Identifying the biophysical origins of the canonical CSD

With the canonical CSD (*Figure 2B*) reproduced, we can use the model to probe the biophysical origins of its sinks and sources. We began by removing all recurrent connections and only feeding the LGN input to the model to find the contribution from the thalamocortical synapses onto excitatory and inhibitory neurons (*Figure 7A*). The main thalamic contribution to the CSD is from synapses onto excitatory neurons, in line with the expectation that neurons with a spatial separation between synaptic input currents and the return currents dominate the cortical LFP generation (*Einevoll et al., 2013*). (Neurons without apical dendrites will have largely overlapping synaptic input currents and return currents, resulting in a cancellation of current sinks and sources.).

We further observed that the early L4 and the sustained L5/L6 sinks are present in the CSD contributions of excitatory neurons, though the magnitude of the L5/L6 sink is substantially reduced compared to its magnitude when the model is configured with recurrent synapses intact (*Figures 4B–6D*). The sustained L2/3 sink and L4 source, on the other hand, were not visible. This suggests that the early L4 sink and the L5/L6 sink are at least partly generated by thalamocortical synapses. However, the substantially diminished magnitude of the L5/L6 sink indicates that recurrent synapses also contribute significantly to the generation of this sink.

We then removed the LGN input and added the feedback (while keeping the recurrent connections cut), which resulted in a prominent upper layer dipole, with the sink residing in L1 and L2/3, and the source residing in L4 (*Figure 7B*). Together with their absence when input came from LGN only (*Figure 7A*), this suggests that the sustained L2/3 sink and the L4 source in the canonical pattern originate at least in part from the feedback synapses onto the apical dendrites of L2/3 and L5 pyramidal cells and the activity this input generates.

To assess the extent to which active channels at the somata contributed to the CSD pattern, we compared the CSD resulting from a simulation with both LGN and feedback input (where the recurrent connections were still cut) when we included or excluded the active channels (NaT, NaP, NaV, h, Kd, Kv2like, Kv3_1, K_T, Im_v2, SK, Ca_HVA, Ca_LVA; only at the soma [see supplementary information in *Gouwens et al., 2018* for definitions]) on all neurons in the model. The most prominent discrepancy between the CSD with and without active channels is the magnitude of the L4 source and the L2/3 sink (*Figure 7C*). In this all-passive setting, the L4 source is significantly attenuated, and the L2/3 sink is either absent or dominated by a source in the same region.

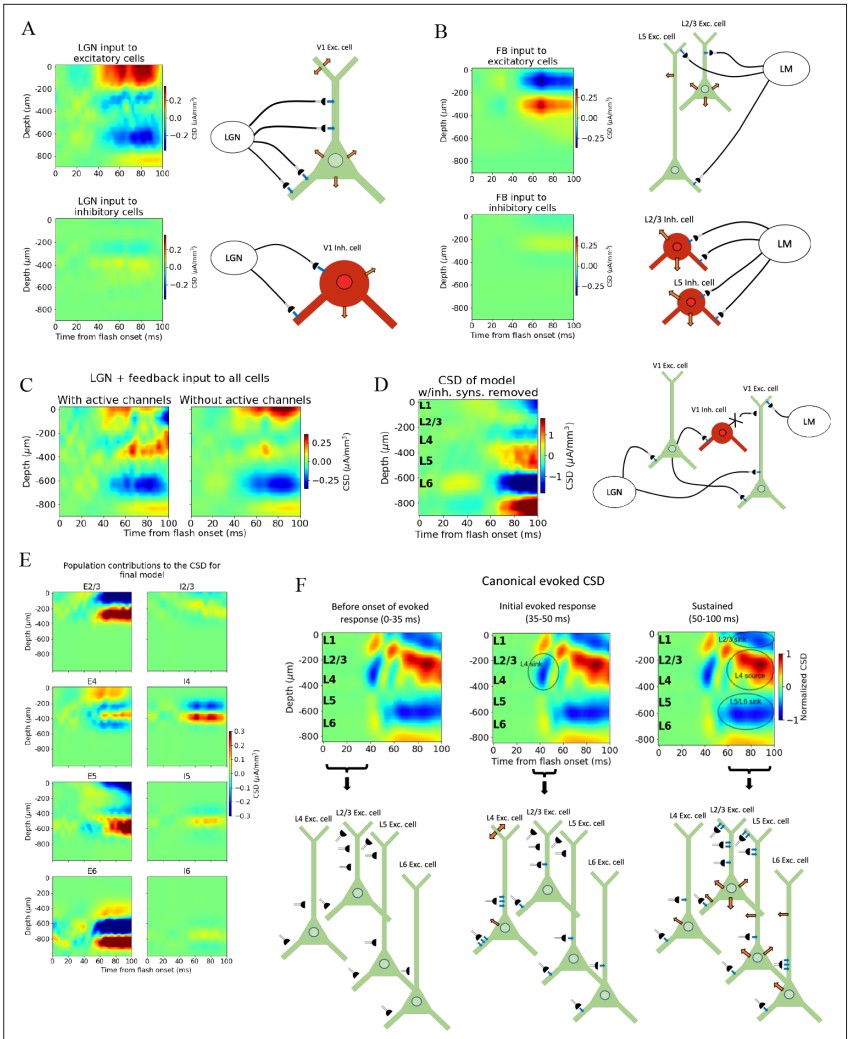

**Figure 7.** Biophysical origin of canonical current source density (CSD). (**A**) Sinks and sources generated from thalamocortical and (**B**) feedback synapses. The schematics illustrate which synapses cause the observed sinks and sources. Blue arrows indicate inflowing current (sinks), while orange arrows indicate outflowing current (sources). (**C**) Total CSD from thalamocortical and feedback synapses (without recurrent connections) with (left) and without (right) active channels in the V1 neurons. (**D**) Total CSD of model with both thalamocortical and feedback input when inhibitory synapses are removed (cross indicates removed connection). (**E**) Population contributions to the total CSD in final model with both lateral geniculate nucleus (LGN) and feedback input and recurrent connections. (**F**) Summary of biophysical origins of the main contributions to the sinks and sources in the canonical CSD in different periods of the first 100 ms after flash onset. More arrows mean more current. Left: before onset of evoked response (0–35 ms). The average inflowing and outflowing current in V1 neurons is zero in this time window. Middle: initial evoked response (35–50 ms). The L4 sink is primarily generated by inflowing current thalamocortical synapses onto L4 excitatory cells. Right: sustained evoked response (50–100 ms). The L5/L6 sink is primarily due to inflowing currents from thalamocortical synapses and recurrent excitatory synapses. Inflowing current at synapses from higher visual areas (HVAs) onto apical tufts of L2/3 and L5 excitatory cells generates, in part, the L2/3 sink, and the resulting return current generates, in part, the L4 source in this time window.

The online version of this article includes the following figure supplement(s) for figure 7:

**Figure supplement 1.** Effects of removing recurrent inhibition on population current source density (CSD) and firing rates.

**Figure supplement 2.** Effect of cell orientation on current source density (CSD) contributions of L4 inhibitory cells.

**Figure supplement 3.** Silencing feedback from latero-medial (LM) in model during evoked response.

We explored whether the contributions from currents in recurrent connections come primarily from excitatory or inhibitory synapses by removing all connections from inhibitory (Pvalb, Sst, Htr3a) neurons to all other neurons, so that all postsynaptic currents stem from excitatory thalamocortical synapses, excitatory synapses from HVAs, or recurrent excitatory synapses in V1 (*Figure 7D* and *Figure 7—figure supplement 1*). Note that inhibitory synaptic currents give rise to sources, while excitatory synaptic currents give rise to a sink. Of course, without inhibition, the network is unbalanced, which limits the conclusions that can be drawn from this simulation. However, the fact that the major sinks and sources are still present is an indication that the currents from excitatory input account for the majority of the sinks and sources observed in the experimental CSD.

The contributions from each population to the total CSD in the final model (*Figure 6D*) with both LGN and feedback input and intact recurrent connections are displayed in *Figure 7D*. From this, it is apparent that the L5/L6 dipole is mainly generated by L6 excitatory cells, the L2/3 sink stems from sinks at the apical tufts of the L2/3 and L5 excitatory cells, the L4 sink from both the L4 excitatory and inhibitory cells, while the L4 source is a mix of sources from mainly L2/3, L4, and L5 excitatory cells, as well as the L4 inhibitory cells. (The magnitude of the CSD contribution from L4 inhibitory cells is greater than anticipated. Given their lack of apical dendrites, we would expect their postsynaptic current sinks and sources to largely cancel [*Einevoll et al., 2013*]. Their contribution can be reduced by scrambling the 3-D orientation of these cells [*Figure 7—figure supplement 2*]. However, we cannot rule out that L4 inhibitory cells can have a contribution comparable in magnitude to the excitatory cells with the data we have available. We therefore let the L4 inhibitory cells keep their original orientation here.)

We investigated what would happen if we turned the feedback off again at 60 ms for the final model version (*Figure 7—figure supplement 3*). Most notably, we found that the sustained L4 source was replaced by a sink, and that the sustained L2/3 sink turned more transient as it was significantly reduced in magnitude after about 70 ms. This CSD pattern serves as a prediction that can be tested experimentally, for example, by optogenetic silencing of the HVAs in the sustained periods of the evoked response (e.g., see *Keller et al., 2020*).

We summarize the main contributions to the canonical CSD in *Figure 7F*. Before the onset of the evoked response (0–35 ms) there is, on average, no significant net inflow or outflow of current to any neurons. Around 40 ms, an inflow of current from excitatory thalamocortical synapses onto all excitatory neurons and all Pvalb inhibitory neurons appears, with the largest current coming from the synapses targeting basal and apical dendrites of L4 excitatory cells. This is the primary origin of the L4 sink. Following this initial L4 sink, there is a sustained sink in L5/L6 arising at ~50 ms, which originates partly from thalamocortical synapses onto L6 excitatory cells and partly from recurrent synapses from excitatory populations in V1 onto L6 excitatory cells. At ~60 ms, a sustained sink emerges in L1 and L2/3, which partly originates in synapses from HVAs targeting apical tufts of L2/3 and L5 excitatory cells. This feedback results in a stronger return current at the soma and basal dendrites of L2/3 excitatory cells and L5 excitatory cells.

## Discussion

In the present study, we analyzed experimentally recorded spikes and LFP during presentation of full-field flashes from a large-scale visual coding dataset derived from mouse visual cortex (*Siegle et al., 2021*) and simulated the same experimental protocol using a biophysically detailed model of mouse V1 (*Billeh et al., 2020*). Our analysis of the experimental data focused on the responses in LGN and cortical visual areas V1 and HVAs. We found that the evoked CSD in V1, computed from the LFP, is captured by a canonical pattern of sinks and sources during the first 100 ms after stimulus onset (*Figure 2B*). This canonical CSD, in response to a flashed, bright field pattern, explains half (50.4 %) of the variance in the trial-averaged CSD responses across animals.

Both the early L4 sink with concurrent sources above and below and the L5/L6 sink with a source below were observed with a similar timing by *Senzai et al., 2019*. The L4 source and L2/3 sink were also observed in that study but emerge somewhat later than in our data – just after 100ms as opposed to ~60 ms in our canonical pattern. This discrepancy in onset might simply be due to differences in stimuli. In *Senzai et al., 2019*, the animals were exposed to 100 ms light pulses, while the animals in our data were presented with 250 ms whole-field flashes of a white screen. Nonetheless, the canonical CSD pattern exhibits good overall agreement with the pattern seen in *Senzai et al., 2019*. Studies in non-human primates where the animals were exposed to flashes also demonstrate good

spatio-temporal agreement with the CSD observed here, with sinks and sources occurring not only in the same layers and in the same order, but also at the same time after stimulus presentation (*Givre et al., 1994*; *Schroeder et al., 1998*).

We introduced the WD as a method to evaluate the difference between two CSD patterns and used it to quantify the variability in trial-averaged CSD between animals (*Figure 2D*), the trial-to-trial variability in CSD within animals (*Figure 2F and G*), and the difference between the model CSD, the trial-averaged CSD of individual animals, and the canonical CSD pattern. This application of the WD to compare CSD patterns comes with certain considerations that are important to note. First, although we compute WD for sinks and sources separately, sinks and sources do not arise independently. Current leaving the extracellular space in one place leads to current entering the extracellular space in another place, so current sinks and sources are inter-dependent. Second, the cost of shifting a sink or a source in space relative to shifting it in time is determined by the relative resolution in space vs. time. This relative cost does not necessarily correspond to the actual cost of changing the underlying physiology such that two distributions of sinks or sources match spatially vs. temporally. Determining the most appropriate relative cost of moving sinks and sources in space vs. time would require more detailed data than currently available and is beyond the scope of this study.

For the firing rate analysis, we utilized KS similarity and correlation to quantify experimental variability and model performance with regard to magnitude and temporal profile, respectively. We also investigated the statistics of unit spike firing by computing the moments of the distributions across cells in each population of peak firing and latencies to peak firing for both individual animals and model versions (*Figure 6—figure supplements 2 and 3*). Systematic use of quantitative metrics for biophysical modeling at this scale is still relatively uncommon, and our work establishes a set of measures for testing the model on LFP and spiking simultaneously, which can be useful for future studies in the field. Of course, there may well be other metrics that are equally or more suitable, and a systematic investigation into what would be the optimal metrics to apply is an important avenue for future work.

Our aim was to simultaneously reproduce experimentally recorded spikes and the CSD in our simulations. The original model captured spiking responses to gratings well (reproducing, e.g., direction selectivity distributions for different neuronal populations) with variable success when applied to other visual stimuli (*Billeh et al., 2020*). It was not originally tested on LFP/CSD. We found that, for the full-field flash stimulus, this model did not reproduce the CSD pattern in the upper layers of V1, and the spiking responses for this stimulus also exhibited a number of discrepancies.

After making selective adjustments to the recurrent synaptic weights, the model could reproduce the experimental firing rates (*Figure 5A–C*), though the discrepancy between the model CSD and the canonical CSD remained (*Figure 5D and E*), with only minor differences relative to the CSD of the original model (*Figure 4B*). The fact that the model can capture the experimental firing rates without capturing the experimental CSD and that adjustments to the synaptic weights yielded significant alterations in firing rates with only small changes in the CSD supports the point that LFP/CSD reflects aspects of circuit dynamics that are complementary to those reflected in locally recorded spikes.

Previous simulation studies demonstrated the importance of synaptic placement in shaping the LFP and CSD signature (*Einevoll et al., 2007*; *Pettersen et al., 2008*; *Lindén et al., 2010*; *Lindén et al., 2011*; *Łęski et al., 2013*; *Hagen et al., 2017*; *Ness et al., 2018*). To uncover the model adjustments that capture firing rates and CSD simultaneously, we explored the effects of changes in the synaptic positioning. In one case, we placed all excitatory synapses onto only basal or apical dendrites of L4 excitatory cells, as opposed to their original placement on both apical and basal dendrites. Moving all excitatory synapses onto basal dendrites resulted in substantial changes in both the pattern and magnitude of the CSD contribution from these L4 excitatory cells, with only minor changes to their firing rates (*Figure 5F and G* and *Figure 5—figure supplement 1*). Placing all excitatory synapses on apical dendrites led to somewhat larger changes in firing rates, though still similar to the firing rate of the original model, and to even bigger changes in the CSD magnitude. Performing the same manipulations of synaptic placement on L2/3 or L5 excitatory cells resulted in a similar decoupling of CSD and firing rates (*Figure 5—figure supplements 3 and 4*). It should, however, be noted that the decoupling is neither perfect nor universal.

This demonstrates a two-way dissociation of the firing rates and the pattern of sinks and sources in the CSD: The firing rates can be substantially altered with small effects on the CSD by adjusting the

synaptic weights, and the CSD can be substantially altered with only small effects on the firing rates by adjusting synaptic placement. These results align with findings in, for example, *Schroeder et al., 1998* and *Leszczyński et al., 2020*, where a lack of correlation between MUA and CSD in the upper layers of V1 during flash exposure to non-human primates suggested that these signals can sometimes be decoupled. Our findings imply that the LFP can reveal deficiencies in the model architecture that would not be evident from the firing rates alone, and that, to a certain extent, models can be optimized for firing rates and CSD independently.

Past studies have suggested that LFP can be modulated by attention through feedback during evoked responses (*Mehta et al., 2000a*; *Mehta et al., 2000b*). However, their findings indicated that V1 was not significantly affected by this modulation until the period 250 ms or later after stimulus onset. A recent study showed that feedback from higher visual areas can in fact exert a strong influence on the magnitude of LFP already at around 80 ms after stimulus onset (*Hartmann et al., 2019*). To investigate whether such cortico-cortical influence can contribute to the sinks and sources in the later stages (>50 ms) of the canonical CSD pattern, we added feedback consisting of experimentally recorded spikes from the higher cortical visual area LM (*Siegle et al., 2021*) impinging on synapses placed onto V1 neurons in our model, using anatomical data (*Glickfeld and Olsen, 2017*; *Marques et al., 2018*; *Hartmann et al., 2019*; *Keller et al., 2020*; *Shen et al., 2020*).

We found that the feedback can play a significant role in shaping the sustained sinks and sources (*Figure 6B–E*). The resulting model CSD reproduced the major sinks and sources identified in the canonical CSD pattern and was no longer among the outliers compared to the experimental variability (*Figure 6D*). Interestingly, absence of the feedback was not apparent from analysis of the firing rates alone, as the firing rates were already within the experimental variability before adding the feedback, further underscoring the utility of the LFP in illuminating structure–function relations in the circuit. Contributions from other visual cortical areas were not included, even though they too impinge upon neurons in V1 (*Harris et al., 2019*; *Siegle et al., 2021*), due to the lack of data characterizing such connections. This awaits future work. Finally, turning off the feedback from LM at 60ms disrupted the CSD pattern in layers 2/3 and 4 during the sustained period (*Figure 7—figure supplement 3*), which serves as a prediction for what will be observed if HVAs, and particularly LM, are silenced in this period of an evoked flash response. Our findings accord with the view that basal dendrites are the main targets for feedforward input, while the tufts of apical dendrites are the main targets for feedback input, even though basal dendrites can also receive input from long-range feedback connections and apical dendrites receive input from feedforward connections (*Aru et al., 2020*).

In our analyses, we aligned the CSD plots to depths obtained from histology and provided in Allen Common Coordinate Framework (CCF) coordinates. An alternative approach is to align the CSD to landmarks in the data (*Senzai et al., 2019*). In *Figure 6—figure supplement 7*, we explored whether utilizing landmarks rather than histology for alignment would affect our results. We found that this approach did not change our conclusion as the final model CSD was still within the experimental variability while the original and intermediate model CSD were outliers both when using the first principal component and when using the plain average as the target (*Figure 6—figure supplements 7B and C and 8*). Lastly, using the plain average rather than the first principal component as the target did not significantly affect our results when we aligned to histology (*Figure 6—figure supplement 9*).

With the major sinks and sources of the canonical CSD pattern reproduced, we explored their biophysical origins. We found that the initial L4 sink originates in the thalamocortical input to L4 excitatory cells, which aligns with suggestions made in *Mitzdorf, 1987*, *Swadlow et al., 2002*, and *Senzai et al., 2019*. The sustained L5/L6 sink comes from postsynaptic currents in L6 excitatory cells triggered by a combination of thalamocortical and recurrent excitatory inputs. The sustained L2/3 sink stems, in part, from input from LM onto the apical tufts of L2/3 and L5 excitatory cells. The sustained L4 source has its origins in a mixture of return currents from L2/3 and L5 excitatory cells resulting from the abovementioned feedback onto the apical dendrites of these cells, as well as contributions from L4 excitatory and inhibitory cells (*Figure 7A, B, D and E*).

In line with observations made by *Reimann et al., 2013*, we found that the somatic voltage-dependent membrane currents significantly shape the CSD signature (*Figures 5H and 7C*). Even so, our findings still emphasize the importance of synaptic inputs in sculpting the CSD, as the addition of synaptic input (*Figure 6A–E*) and changes to synaptic placement (*Figure 5F*) substantially altered the CSD pattern.

Recent investigations into the unitary LFP (the LFP generated by a single neuron) from inhibitory and excitatory synapses have suggested that inhibitory inputs exert a greater influence on LFP than excitatory input (*Bazelot et al., 2010*; *Teleńczuk et al., 2017*; *Telenczuk et al., 2020*). While this may be true of unitary effects, the total effect of excitatory input can still be greater if there are significantly more excitatory than inhibitory cells, and, correspondingly, significantly more excitatory synapses. In this V1 model, inhibitory cells make up about 15% and excitatory cells about 85% of the total number of cells, reflecting the cellular composition in mouse V1. Whether the pronounced dominance of excitatory cells is enough to make up for the reduced unitary influence of excitatory cells is an interesting area for further research.

This investigation into the biophysical origins of sinks and sources is limited by the fact that the contributions from recurrent connections are difficult to estimate precisely due to the nonlinear effects of these connections. That is, their contribution cannot simply be found by subtracting the CSD from thalamocortical and feedback synapses with all recurrent connections removed (*Figure 7A and B*) from the total CSD with the same input and recurrent connections intact (*Figure 6C*, right). Still, this analysis provides an initial estimate into the biophysical origins of the sinks and sources observed experimentally and demonstrates the insights that can be obtained from modeling of extracellular signals.

There is ample evidence that firing rates and LFP are modulated by the behavioral state of the animal, including measures like the pupil size (considered to be a proxy for arousal level) or running speed (*Niell and Stryker, 2010*; *McGinley et al., 2015*; *Vinck et al., 2015*; *Saleem et al., 2017*). In this study, the responses averaged over all trials have been the target for the modeling, without regard to any state-dependence of the responses. Our understanding of the state-dependent responses could benefit from the potential to probe the biophysical origins of extracellular signals. Therefore, reproducing these state-dependent responses is an important avenue for future research.

Note that the set of synaptic weights and other parameters that can reproduce the experimental firing rates and CSD is unlikely to be unique. This is a consequence of the degeneracy inherent to biological neural networks, known from both simulation and experimental studies, as many different parameterizations of neuronal networks can perform the same functions (*Prinz et al., 2004*; *Marder and Goaillard, 2006*; *Drion et al., 2015*; *O'Leary, 2018*). Thus, our network should only be considered an example of a circuit model that can produce firing rates and CSD that match the experimental observations. Obtaining multiple solutions and characterizing their diversity using automatic searches of the parameter space will be an interesting direction for future work. We did not utilize such an approach here because the number of simulations required (typically, many thousands or more for automatic optimization approaches) would currently be prohibitively expensive on a model of this scale and level of complexity: running a 1 s simulation with this model takes ~90 min on 384 CPU-cores (*Billeh et al., 2020*); a single trial in this study simulates 0.75 s of activity.

The original model used as a starting point here produced firing rates and direction and orientation tuning consistent with recordings during presentations of drifting gratings (*Arkhipov et al., 2018*; *Billeh et al., 2020*). In this study, we focused on the analysis and modeling of the response to full-field flashes, but when the final model was tested with the drifting gratings stimulus utilized in *Billeh et al., 2020*, we found that the present, revised model continued to exhibit orientation and direction tuning even though the adjustments were made with the aim to reproduce the observed CSD and population firing rates during full-field flashes (*Figure 6—figure supplement 6*). Ideally, the model should reproduce both firing rates and LFP simultaneously not only for flashes or drifting gratings, but for any visual stimulus (out-class generalization). This is a long-term goal and can be called 'the holy grail' of visual system modeling.

In this study, we developed a systematic framework to quantify experimental variability in both LFP/CSD and spikes and to evaluate model performance. We identified a canonical CSD pattern observed during presentations of full-field flash stimuli and obtained a bio-realistic model that reproduced both the canonical CSD pattern and spikes simultaneously. This model thus reproduces, in a quantitative manner, the shape and timing of current sinks and sources observed experimentally. We utilized this validated model to explain, mechanistically, the biophysical origins of the various current sinks and sources and their location across the layers of visual cortex. Our models are freely shared and should be useful for future studies disentangling the mechanisms underlying spiking dynamics and electrogenesis in the cortex.

## Materials and methods

### Experiments

#### Quality control

Of the 58 mice in the visual coding dataset, 9 were excluded because the exact probe location could not be recovered due to fading of fluorescent dye or artifacts in the OPT volume (*Siegle et al., 2021*). Another five animals were excluded because they were missing LFP recordings from V1 during presentation of the flash stimulus. Thus, data for 44 animals were retained for the CSD analysis.

For the spike analysis, the same nine animals for which the exact probe location could not be recovered were excluded, and two additional animals were excluded because they did not have any cells recorded in V1, leaving a total of 47 animals for this part of the data analysis.

#### Neuronal classification

We distinguished between RS and FS cells by the time from trough to peak of the spike waveforms (*Barthó et al., 2004*). For cortical cells, the spike duration was bimodally distributed with a dip at ~0.4 ms, while for thalamic cells, it was bimodally distributed with a dip at ~0.3 ms (*Figure 3—figure supplement 1*). Thus, the cutoff in the classification of cells as RS or FS was set at 0.4 ms for LM and V1, and at 0.3 ms for cells in LGN. Note that several studies have demonstrated that some pyramidal neurons may have spike waveforms short enough to be classified as FS cells (*Vigneswaran et al., 2011*; *Lemon et al., 2021*). Thus, some caution is warranted when interpreting the population firing rates.

When comparing the model firing rates to the experimental firing rates, the excitatory and non-Pvalb populations were grouped together in each layer of the model to make up the RS cells in L2/3, L4, L5, and L6, while the Pvalb cells across all layers were grouped together to make up the FS cells of V1. The layer boundaries were taken from the Allen CCF (*Oh et al., 2014*), allowing for the assignment of each neuron's position to a specific cortical layer (*Siegle et al., 2021*).

### Model

The model consists of both biophysically detailed multicompartment neurons and leaky-integrate-and-fire (LIF) point-neurons. In total, there are 51,978 multicompartment neurons with Hodgkin–Huxley conductances at the soma and only passive conductances at the dendrites. These are arranged in a cylinder of radius of 400 µm and height 860 µm (corresponding to the average cortical thickness of V1 taken from the Allen CCF; *Billeh et al., 2020*; *Oh et al., 2014*). This cylinder makes up the 'core' of the model and is surrounded by an annulus of 178,946 are LIF neurons which has the same height and a thickness of 445 µm. This makes the total number of neurons in the model 230,924 and the radius of the whole cylinder with both biophysically detailed and LIF neurons 845 µm. There are 17 different classes of neuron models. In each layer from 2/3 to 6, there are one excitatory and three inhibitory classes (Pvalb, Sst, Htr3a), while in layer 1 there is a single Htr3a class. The LGN module providing thalamocortical input to the model consists of 17,400 units selectively connected to the excitatory neurons and Pvalb neurons in L2/3 to L6, as well as the non-Pvalb neurons in L1. The background input to all neurons in the model comes from a single Poisson source firing at 1 kHz and represents influence from the rest of the brain. The feedback input to L2/3 and L5 excitatory, Pvalb, and Sst neurons comes from a node representing LM.

### Simulation configuration

Instructions on how to run simulations of the model are provided in *Billeh et al., 2020*. The files and code necessary to run the model versions presented in *Figures 4–6* are provided in the directories old_model_fig4 intermediate_model_fig5, and final_model_fig6, respectively, on Dryad (see 'Data availability').

### Data processing

#### LFP and CSD

The LFP in simulations was obtained from the extracellular potential by first downsampling to every other electrode along the probe (resulting in a spatial separation of 40 µm between each recording electrode, equal to the spacing in the public Neuropixels data) and using a low-pass fifth-order

Butterworth filter with a cutoff frequency of 500 Hz (utilizing functions scipy.signal.butter and scipy.signal.filtfilt). The same filtering was applied to get the experimental LFP. The CSD was calculated from the experimental and model LFP using the delta iCSD method (*Pettersen et al., 2006*), where the radius of laterally (orthogonal to the probe axis) constant CSD was assumed to be 400 μm – the radius of the V1 model's 'core' region consisting of biophysically detailed multicompartment neurons. For the experimental CSD, this radius was set to 800 μm, roughly corresponding to the size of mouse V1.

## Visual stimulus

The stimulus used to compare the model and the experiments was full-field flashes. In the experiments, the mice were presented with gray screens for 1 s, followed by 250 ms of white screen, and then 750 ms of gray screen over 75 trials. In the simulations, both the stimulus presentation and the pre- and the poststimulus gray screen periods lasted 250 ms, and the number of trials was 10.

### Input from LGN)

Originally, the LGN spike trains used as input to the model were generated with the FilterNet module provided with the model, using 17,400 'LGN units' (*Billeh et al., 2020*). However, when this input was used for simulations, the onset of the evoked response in V1 was 20–30 ms delayed in comparison with experiments. Therefore, we used experimentally recorded LGN spike trains as input to the model instead. We assigned a recorded spike train to each of the 17,400 LGN units in all trials. In total, the public Neuropixels data contain recordings from 1263 regular-spiking LGN neurons across 32 animals during 75 trials of full-field flash presentations. We divided the total pool of spike trains into 10 subsets, and then randomly sampled spike trains from one subset in each trial until all 17,400 LGN units had been assigned a spike train in all trials.

### Input from LM

The experimentally recorded spike trains in the LM were used to construct the feedback input to V1. In total, the public Neuropixels data contain recordings from 1823 RS LM neurons across 42 animals during presentations of the full-field flash stimulus. Spikes were randomly sampled from the pool of all spike trains to construct a spike train that was used as input to all the cells that were targeted by the feedback in the model. All neurons received the same spike train.

### Background input

The input from the Poisson source firing at 1 kHz was not stimulus dependent. It is a coarse representation of the continuous influence of the rest of the brain on V1.

## Dendritic targeting

The rules for placement of synapses were set in *Billeh et al., 2020* and were based on reviews of literature on anatomy.

### LGN to V1

In the original model, the synapses from LGN onto excitatory V1 neurons were placed on apical and basal dendrites within 150 μm from the soma, while synapses onto inhibitory V1 neurons were placed on their soma and on their basal dendrites without distance limitations (*Billeh et al., 2020*). This placement was left unchanged in this study.

### V1-V1

The synapses for recurrent connections were placed according to the following rules in the original model (*Billeh et al., 2020*):

#### Excitatory-to-excitatory connections

All synapses from excitatory V1 neurons onto other excitatory V1 neurons were placed along the dendrites and avoided the soma. In layers 2/3 and 4, the placement of synapses was restricted to be

within 200 μm from the somata, while in layers 5 and 6, they could be placed anywhere along the dendrites.

### Excitatory-to-inhibitory connections
All synapses from excitatory V1 neurons onto inhibitory V1 neurons were placed on their somata or dendrites without any distance limitations.

### Inhibitory-to-excitatory connections
Synapses from Pvalb neurons onto excitatory V1 neurons were placed on the soma and on the dendrites within 50 μm from the soma. Synapses from Sst neurons were placed only on dendrites and only more than 50 μm from the soma. Synapses from Htr3a neurons were placed on dendrites between 50 and 300 μm from the soma.

### Inhibitory-to-inhibitory connections
Synapses from inhibitory neurons to other inhibitory neurons were placed according to the same rules as the inhibitory-to-excitatory connections described above.

These placement rules were kept in this study, except for the synapses from excitatory neurons to excitatory L6 neurons. Here, they were restricted to be within 150 μm of the soma. The purpose of this restriction was to reduce the spatial separation between the current sink and source, and thereby decrease the magnitude of the L6 sink-source dipole.

### LM-V1
The synapses from the node representing LM to V1 were placed on the apical dendrites of L2/3 neurons (within 150 μm from the soma), on the apical tufts (>300 μm from the soma) and the basal dendrites (within 150 μm from the soma) of L5 excitatory cells, and on the somata and basal dendrites of L2/3 and L5 inhibitory cells (at any distance from the soma).

## Adjusting synaptic weights
In the original model, the synaptic weights of thalamocortical connections were based on experimental recordings of synaptic current, while the synaptic weights for recurrent connections were initially set to estimates from literature, then optimized to a drifting gratings stimulus until the model reproduced experimental values of orientation and direction selectivity. In this study, the synaptic weights for thalamocortical connections were left unchanged from the original model. Before the addition of feedback from higher visual areas to the model, the synaptic weights for recurrent connections in V1 were multiplied by factors in the range [0.2, 2.5].

In the original model, the input from the background node represented the influence of the rest of the brain on V1, which included the influence from higher visual areas such as LM. This means that some of the feedback influence from LM on V1 should be present (though coarsely represented) in the input from the background node. When the influence of input from LM to feedback-targeted cells is modeled on its own, the influence of the background node must be updated accordingly. Thus, after the addition of feedback, the synaptic weights from the background node to the populations targeted by feedback (the L2/3 and L5 excitatory, Pvalb, and Sst cells) were multiplied by factors in the range [0.2, 0.5]. The synaptic weights from the node representing LM were initially set equal to the original weights between the background node and the populations targeted by the feedback, but this led to too high firing rates compared to the experimental firing rates in these populations, so they were multiplied by factors in the range [0.2, 0.5]. Finally, the connections from Pvalb neurons in V1 to L2/3 excitatory neurons and L5 excitatory cells were re-scaled in the range [0.8, 1.2] times the weights set prior to the addition of feedback. The ranges reported here were set after experimenting with different ranges to find what would allow the model to reproduce the experimental observations. Only a single value within each range was used in the final model.

## Quantification and statistical analysis
### Firing rates
The time-resolved population firing rates (bin size 1 ms, filtered using scipy.ndimage.gaussian_filter with sigma = 2) were computed by averaging the spike count over all cells in a population and over all

trials (10 trials in the simulations and 75 trials in the experiments). The distribution of firing rates across cells used in the calculation of the KS similarities was computed by averaging over the time windows baseline, initial peak, and sustained activity (defined in *Figure 3*) and over all trials.

## Kolmogorov–Smirnov similarity

The KS similarity scores (*Billeh et al., 2020*) were computed by first calculating the KS distance (using the function scipy.stats.ks_2samp) between two distributions of firing rates across cells, and subtracting this number from 1, such that a KS similarity score of 1 implies identity and a score of 0 implies no overlap between the two distributions. In the comparison of the model to the experimental data, the KS similarity was computed between the distribution of firing rates across cells in each RS and the FS population of the model and the distribution of firing rates across cells from all animals in the corresponding populations. To assess the variability in the experiments, the KS similarity was calculated between the distribution of firing rates across cells in the same RS and FS populations in individual animals, provided there were more than 10 cells recorded in a given population in this animal, and the distribution of firing rates across cells from all other animals.

## Correlation

We computed the similarity in the profile of time-resolved population firing rates with the Pearson correlation coefficient (using the function scipy.stats.pearsonr). The correlation between the model and the experimental firing rates was calculated between model population firing rates and the population firing rates averaged across cells from all animals. The level of experimental variability was assessed by calculating the correlation between population firing rates in each animal and the population firing rates averaged across cells from all other animals.

## CSD analysis

Since the number of recording electrodes in V1 are not the same in all animals, we interpolated the CSD of each animal and the CSD from simulations onto dimensions of the same lengths ($M = 30$ points along the depth and $K = 100$ points along the time axis for 100 ms time windows) before we quantitatively analyzed the CSD.

### PCA

The trial-averaged CSD of each animal was flattened into a vector of length $M \times K = 3000$, and the vectors of all $N = 44$ animals were stacked together into a matrix of size $44 \times 3000$. Then, we performed PCA (using sklearn.decomposition.PCA) on this matrix to obtain the principal components that would constitute sums of weighted contributions of the trial-averaged CSD patterns.

### Wasserstein distance (WD)

The first Wasserstein distance $W\left(P_1, P_2\right)$ between two distributions $P_1$ and $P_2$ is defined as

$$W\left(P_1, P_2\right) = \inf_{\gamma \in \Gamma\left(P_1, P_2\right)} \int c\left(x, y\right) \gamma\left(x, y\right) dx dy$$

where $c\left(x, y\right)$ is the cost of moving a unit 'mass' from position $x$ to $y$ following the optimal transport plan $\gamma\left(x, y\right)$ in all transport plans $\Gamma\left(P_1, P_2\right)$ (*Rubner et al., 1998*; *Arjovsky et al., 2017*).

In the utilization of WD to quantify the similarity between two CSD patterns, the distance between the distribution of sinks in the two patterns $W\left(P_{sinks, 1}, P_{sinks,2}\right)$ and the distance between distribution of sources of the two patterns $W\left(P_{sources, 1}, P_{sources,2}\right)$ are calculated separately and summed to form a total WD between the two CSD patterns:

$$W_{CSD}\left(P_1, P_2\right) = W\left(P_{sinks,1}, P_{sinks,2}\right) + W\left(P_{sources,1}, P_{sources,2}\right)$$

where $P_1$ and $P_2$ refer to the two CSD patterns. The Python Optimal transport library (https://pythonot.github.io/index.html) was used to implement this calculation.

## Acknowledgements

Research reported in this publication was supported the Simula School of Research and Innovation, CINPLA, the European Union Horizon 2020 Research and Innovation Program under Grant Agreement No. 785907 and No. 945539 (Human Brain Project [HBP] SGA2 and SGA3), the National Institute of Neurological Disorders and Stroke of the National Institutes of Health under Award Number R01NS122742, and by the National Institute of Biomedical Imaging and Bioengineering of the National Institutes of Health under Award Number R01EB029813, as well as by the Allen Institute. The content is solely the responsibility of the authors and does not necessarily represent the official views of the National Institutes of Health. We acknowledge the use of Fenix Infrastructure resources, which are partially funded from the European Union's Horizon 2020 research and innovation program through the ICEI project under the grant agreement No. 800858. We thank the Allen Institute founder, Paul G Allen, for his vision, encouragement, and support.

## Additional information

### Competing interests

Christof Koch: holds an executive position, and has financial interest, in Intrinsic Powers, Inc, a company whose purpose is to develop a device that can be used in the clinic to assess the presence and absence of consciousness in patients. This does not pose any conflict of interest with regard to the work undertaken for this publication. The other authors declare that no competing interests exist.

### Funding

| Funder | Grant reference number | Author |
|---|---|---|
| Simula School of Research | | Atle E Rimehaug |
| European Union Horizon 2020 Research and Innovation program | 785907 | Espen Hagen |
| European Union Horizon 2020 Research and Innovation program | 945539 | Espen Hagen |
| Research Council of Norway | COBRA - project number 250128 | Alexander J Stasik |
| IKTPLUSS-IKT and Digital Innovation | 300504 | Alexander J Stasik |
| National Institute of Neurological Disorders and Stroke | R01NS122742 | Kael Dai<br>Josh H Siegle<br>Shawn R Olsen<br>Christof Koch<br>Anton Arkhipov<br>Yazan N Billeh |
| National Institute of Biomedical Imaging and Bioengineering | R01EB029813 | Kael Dai<br>Josh H Siegle<br>Shawn R Olsen<br>Christof Koch<br>Anton Arkhipov<br>Yazan N Billeh |
| Allen Institute | | Kael Dai<br>Josh H Siegle<br>Shawn R Olsen<br>Christof Koch<br>Anton Arkhipov<br>Yazan N Billeh |

The funders had no role in study design, data collection and interpretation, or the decision to submit the work for publication.

## Author contributions
Atle E Rimehaug, Conceptualization, Software, Formal analysis, Investigation, Visualization, Methodology, Writing – original draft, Writing – review and editing; Alexander J Stasik, Conceptualization, Software, Formal analysis, Investigation, Visualization, Methodology; Espen Hagen, Conceptualization, Resources, Software, Formal analysis, Investigation, Methodology, Writing – review and editing; Yazan N Billeh, Conceptualization, Software, Methodology; Josh H Siegle, Data curation, Investigation, Visualization, Writing – review and editing; Kael Dai, Resources, Software, Methodology; Shawn R Olsen, Resources, Supervision; Christof Koch, Gaute T Einevoll, Anton Arkhipov, Conceptualization, Resources, Supervision, Funding acquisition, Methodology, Project administration, Writing – review and editing

## Author ORCIDs
Atle E Rimehaug ⓘ https://orcid.org/0000-0002-8312-9875
Alexander J Stasik ⓘ https://orcid.org/0000-0003-1646-2472
Josh H Siegle ⓘ https://orcid.org/0000-0002-7736-4844
Shawn R Olsen ⓘ https://orcid.org/0000-0002-9568-7057
Gaute T Einevoll ⓘ https://orcid.org/0000-0002-5425-5012
Anton Arkhipov ⓘ https://orcid.org/0000-0003-1106-8310

## Decision letter and Author response
Decision letter https://doi.org/10.7554/eLife.87169.sa1
Author response https://doi.org/10.7554/eLife.87169.sa2

## Additional files

### Supplementary files
• MDAR checklist

### Data availability
The files necessary to run simulations of the different model versions presented in the paper as well as data resulting from simulations of those model versions are publicly available in Dryad: https://doi.org/10.5061/dryad.k3j9kd5b8. The experimental data set utilized is publicly available at: https://portal.brain-map.org/explore/circuits/visual-coding-neuropixels. The code generated for data analysis and producing the figures in this manuscript is publicly available at: https://github.com/atleer/CINPLA_Allen_V1_analysis.git (copy archived at *Rimehaug, 2023*).

The following dataset was generated:

| Author(s) | Year | Dataset title | Dataset URL | Database and Identifier |
| --- | --- | --- | --- | --- |
| Rimehaug AR, Stasik AJ, Hagen E, Billeh YN, Siegle JH, Dai K, Olsen SR, Koch C, Einevoll G, Arkhipov A | 2022 | Uncovering circuit mechanisms of current sinks and sources with biophysical simulations of primary visual cortex | https://dx.doi.org/10.5061/dryad.k3j9kd5b8 | Dryad Digital Repository, 10.5061/dryad.k3j9kd5b8 |

The following previously published dataset was used:

| Author(s) | Year | Dataset title | Dataset URL | Database and Identifier |
| --- | --- | --- | --- | --- |
| Siegle JH, Wakeman W, Jia X, Heller G, Ramirez T, Graddis N, Mei N, Durand S | 2020 | 20191003_AIBS_mouse_ecephys_brain_observatory_1_1 | https://dandiarchive.org/dandiset/000021 | DANDI Archive 000021, 000021 |

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

## Appendix 1

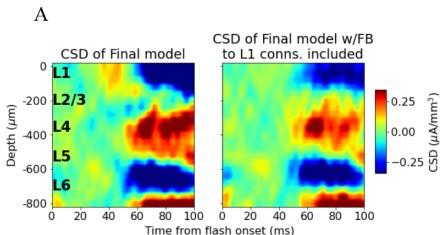

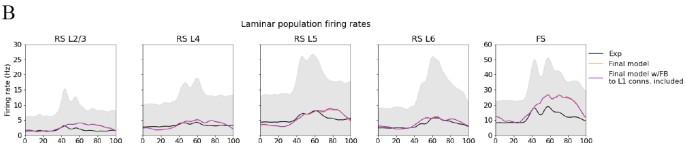

**Appendix 1—figure 1.** Effect of adding feedback connections from latero-medial (LM) to L1 inhibitory cells. (**A**) Left: current source density (CSD) from simulation of final model version without connections from LM to L1 inhibitory cells. Right: CSD from simulation of final model version with connections from LM to L1 inhibitory cells included. (**B**) Population firing rates of experiments (black line), final model without connections between LM and L1 inhibitory cells (orange line), final model with connections between LM and L1 inhibitory cells included (purple colored line).

