## [Editor Report]

The study demonstrates that utilizing the LFP and/or the CSD in modeling can facilitate model configuration and implementation by revealing discrepancies between models and experiments. The analysis of the biophysical origin of the canonical CSD using the model is an interesting and worthy line of investigation. The dissection of CSD components is detailed and exhaustive. A key novelty of this article is the addition of CSD patterns as another constraint to more accurately infer the model parameters beyond its prior state.

---

## [Decision Letter]

**Decision letter after peer review:**

[Editors’ note: the authors submitted for reconsideration following the decision after peer review. What follows is the decision letter after the first round of review.]

Thank you for submitting the paper "Uncovering circuit mechanisms of current sinks and sources with biophysical simulations of primary visual cortex" for consideration by *eLife*. Your article has been reviewed by 3 peer reviewers, and the evaluation has been overseen by a Reviewing Editor and a Senior Editor. The following individuals involved in review of your submission have agreed to reveal their identity: Antonio Fernandez-Ruiz (Reviewer #1); Stephanie R Jones (Reviewer #3).

*Reviewer #1 (Recommendations for the authors):*

Rimehaug et al., addressed two important questions in this work: the contribution of different circuit components to the generation of network evoked responses and the relationship between single cell firing rates and mesoscopic currents. They used a rigorous quantitative approach combining analysis of experimental data and simulations to address these questions in the mouse visual primary visual cortex. The authors start by characterizing the stereotypical V1 laminar response to full-filed light flashes, which replicates previous studies. They then perform a detailed quantification of within and across animal variability of both mesoscopic and single unit firing rate responses. Next, they attempt to reproduce these responses in a previously published biophysical network model that was shown to reproduce realistic orientation and direction tuning cell response. However, they found that the model did not accurately reproduced responses at the level of local field potentials (LFP) and current source density (CSD), so they proceeded to systematically vary model parameters until obtaining a better fit. One of the main findings of the article is that varying the strength of some synapses in model had a strong effect on firing rates but small on the CSD while varying synapse dendritic placement had strong effects on the CSD but small on firing rates. In the last part of the paper, the author use the fully tuned model to provide mechanistic insights into the biophysical basis of the different component of CSD evoked responses.

The results reported in this manuscript and the methods employed are sound, detailed and rigorous and accurately support the authors interpretations. I found of special relevance the demonstration of the importance of incorporating LFP/CSD features to both understand network dynamics a constrain models. I don't have any major concern with this manuscript but suggest below some points where it can be improved.

– Some of the analysis and simulations can be better motivated. For example, it is not clear, in the section starting in line 244, why it is important to quantify CSD variability and how that relates to the broader goals of the study. The manuscript in quite dense and in occasions is not easy to follow how certain results contribute to the main objective. It would help to state these explicitly at the outset and related each subsequent section to them.

– While the dissection of CSD components is detailed and exhaustive, the analysis of unit firing response fall short. In previous works (like Senzai et al., 2018) a stereotypical timing of single unit firign to this type of stimuli is described in addition to the CSD responses. In the paper there are Can the author's model reproduce this timing of unit activation? In addition to reporting average firing rates, the authors should also describe and interpret the temporal dynamics of unit firing and relate it to their mechanistic description of CSD components in the last section.

– When discussing the biophysical mechanisms of CSD generation a useful concept to introduce is that of dipolar moment. Components with a larger dipolar moment (more separation between sink and source) generate larger LFPs.

– To appeal to a broader audience, it would be useful to related the present findings with both, other mesoscopic network patterns in mouse V1 and in different cortical areas (e.g. sensory evoked γ oscillations, UP-DOWN states, etc.)

The authors should do a better job with citations. Most citations about broad topics like LFP biophysics and modelling or separation and characterization of the generating mechanisms of LFP/CSD components are from the same set of authors, that include those of the present paper. Many other groups have also produced relevant work in these topics, even if in different brain structures (as the hippocampus) or animal models (NHPs) but are not cited here.

*Reviewer #2 (Recommendations for the authors):*

Using a biophysically detailed model of mouse primary visual cortex, the evoked response to a full-field bright flash is simulated and compared with experimental data.

This paper aims to show how local field potential signal can be helpful in characterizing neural circuitry, and how it provides complementary information to the analysis of neural firing. Spiking patterns alone, they argue, do not uniquely characterize a neural circuit, i.e. there are many models (or variations to the same model) that are capable of reproducing the spiking activity of data, so requiring the model to simultaneously reproduce the CSD would further restrict the model.

The authors try to demonstrate this point by showing that there is a two-way dissociation between CSD pattern and the firing rate pattern. By adjusting synaptic weights they can reproduce the experimental firing rates statistics with minor changes in CSD and by adjusting synaptic placement they can alter the CSD patterns with minor changes in firing rates statistics.

Another support for this claim is that authors can better match both CSD patterns and firing rates by adding input that simulates a feedback from higher visual cortical areas. When feedback is incorporated in the model the performance must improve and it indeed does so based on CSD and firing rate statistics.

In the final section, after having a well-performing model, authors try to identify the pre and postsynaptic populations responsible for different prominent sinks and sources in evoked CSD pattern. They do this by manipulating different synaptic connections in the model: LGN input, cortical top-down feedback, excitatory recurrent connections onto different target neural populations. By comparing the CSD with only one pathway intact and measuring the contribution of that component to the total CSD pattern the authors qualitatively demonstrate contribution of these input-target components to the CSD.

The main approach chosen in the paper – linking detailed experimental data characterizing the evoked response in V1 (firing patterns of a large number of neurons across layers and circuits and LFP/CSD patterns) and a very detailed model of the circuit is a highly ambitious goal. This reviewer is not sharing the authors' optimism that it is achievable at this point beyond illustrations of how different elements of the model could potentially contribute to some elements of the experimental data. The key novelty of this manuscript compared to a previous paper (Billeh et al. 2020) is adding CSD patterns as an additional constraint to more accurately infer the model parameters beyond its prior state. While it is true that adding constraints would undoubtedly be helpful, there is no guarantee that specific hyperparameters used, components of the model simplified or ignored and gross aggregated measures chosen for comparing the patterns in the data and produced by the model would allow identifiability of the model and use it to explain the experimental data. Authors make no justification for many specific choices for a number of unknown high-dimensional parameters e.g. morphology of particular cell types (e.g. L4 pyramids have minimal basal dendrites, Scala et al. 2019, which has a strong effect on Figure 5 results), somato-dendritic distribution of synaptic inputs from different afferents, choice of interneurons (morphology, intrinsic properties, and connectivity) to include and how to wire them in recurrent and feedforward circuit, etc. Given the nonlinear nature of the network dynamics driven by the thalamic input and its nontrivial dependence on very high-dimensional parameter space, I see no way to identify a true model in that space based on very coarse metrics of aggregated neural activity. Care for realism in the model and the use of real data provides a false impression that the outcome of the effort should be the identification of the actual unobserved parameters of the system and the identification of the model. I think that it is a false impression, that might be misleading the readers.

The choice of metrics and dimensionality reduction approaches used to capture the group results on the CSD and firing rate is not suitable for validating the model or might provide poorly interpretable results. The authors note in the discussion, that there might be more suitable metrics and promise to work on it in the future. However, if the choice of metric used in the present paper makes interpretation questionable it puts in question the validity of all the results and conclusions in the manuscript, except for the actual data and model it is based on. Thus, I find it critical that the analysis methods used are chosen carefully and shortcomings discussed below are taken into account.

Inter-animal CSD alignment. It is obvious from the examples and prior work (Senzai et al. 2019) that variation of the location and scale of many CSD dipoles varies greatly. Any further analysis that relies on the CSD has to be thus rescaled and realigned, as Senzai et al. paper did (see their Figure 1). This would allow capturing the true variability due to physiological structure across all the data and not the variability related to the anatomical location of the electrode. I believe that adding this step would very strongly improve the remaining analysis (see more below).

Canonical CSD pattern. The choice of the first principal component of trial-averaged CSDs as the canonical CSD pattern is not at all obvious and has potential ramifications for the remaining analysis that it is based on. The first principal component is capturing the direction of inter-animal variability in the CSD pattern time-depth space affected by the angle of the probe, alignment of the recording sites array to cortical layers, exact retinotopic location, variation of brain states across individual recording sessions etc. This first eigenvector of the covariance matrix does not necessarily capture a common underlying pattern. In fact, mathematically it is not simply a weighted average but depends on the structure of the remaining unexplained variance (ca 50%). The components of the evoked response that strongly covary would contribute more to the first PC, but those that are uncorrelated or partially independent would contribute in an unpredictable way. In light of the focus of the paper on explaining this multi-component structure, it appears that starting from the decomposition of each animal's evoked response, extracting parameters (time, strength) of individual dipole components and characterizing their variability in population would be the most accurate way of characterizing experimental data.

CSD Distance Metrics. The Wasserstein distance used as a measure of distance between two CSD patterns is sensitive to misalignment of the CSD patterns (see above). The WD statistics cannot differentiate the contribution of the single sink and source components as they are aggregated into a single compound measure to which both anatomical variability contributes potentially more than the physiological variability. At the same time, the key elements of the paper are critically linked to this measure and its ability to capture the similarity between the experiment and the model.

KS-similarity. This statistic captures coarse (max of CDF difference) differences of the CDFs and is used to quantify the variability of firing rate magnitude across different populations. As a distance between two distributions, KS distance is not capturing the shape of the distributions and it's not quantifying where they are different. One can see that KS-similarity score is close to 0.75 for all populations in the dataset – the specificity of this measure is low. Importantly, this distribution similarity measure is not sensitive to the nature of change (direction), affected by the emergence of skew or modes in the distributions etc. Critically, the application of the measure to distinct layers doesn't capture joint spatial patterns of the firing rate across layers, which could get it closer to the CSD similarity measure. As the direction of the rate change is not distinguished, very different spatial patterns of firing rate change would be associated with similar KS-similarity values.

Pearson correlation is used to quantify the similarity in the average temporal profile of firing rates of layer-specific populations in experimental and model data. First, In contrast to the KS-similarity measures that strive to capture the distribution of the population firing rates, but ignores temporal dynamics, this measure ignores the variability of neural firing rates and their temporal dynamics. Changes in the shape of the distribution of firing rates will translate into unpredictable changes of the mean firing rate. Variable temporal profiles of neurons would further contribute to the poor interpretability of the mean firing rate evoked response. Given the generally log-normal distribution of the firing rates in many cortical circuits the mean is poorly describing the distribution. Overall, I can't see how mean rate response could be used as an interpretable statistics characterizing complex spatio-temporal dynamics across populations of neurons. A huge spread of confidence interval in all figures showing mean population PSTH is a testament to this point. Second, as the mean firing rate statistics is unlikely normally distributed, given the nonstationary changes of the firing rate, and might not be even unimodally distributed in the 0-100ms time window, using Pearson correlation coefficient is not ideal as it best measures the linear relationship between two normally distributed random variables. It is deviations from a bivariate normal distribution that would influence the value of correlation coefficient introducing a bias. It is hard to interpret what values ranging between 0.3 to 0.6 reflect if anything. Multiple spectral components of the evoked response would contribute in an unaccountable way to the measure further complicating the interpretation.

A small number of simulated trials yields very noisy (high variance) average CSD and PSTH. Given the importance for the interpretation of the results presented in the paper and the stochastic nature of the simulated data, it would be valuable to run longer simulations (more trials) to reach reliable low-variance estimates of the CSD/PSTH.

Analysis of the biophysical origin of the canonical CSD using the model is an interesting and worthy line of investigation. However, from the approach used in this paper that compares CSD due to one pathway to that generated by the full model, it is difficult to make conclusions about the exclusive contribution of that pathway, since this system with many recurrent connections is nonlinear. Figure 7 only shows single examples and relies on very specific, and somewhat arbitrary parameters that are critical for the outcome. Somato-dendritic distribution and density of synapses between a particular input and target neuron, the morphology of that neuron, and the strength of input all make a large impact on the resulting CSD. Most statements are based on a single example (i.e. specific parameters choice) and eye-balling, with no quantification provided. While it could provide some qualitative intuition for a plausible generative model of the CSD of the evoked response in V1, the large number of unknown parameters that were specifically chosen in the model makes it impossible to generalize the results of these single examples to the experimental data.

An appraisal of whether the authors achieved their aims, and whether the results support their conclusions.

Two-way dissociation between neuron's firing rate and CSD

The two-way dissociation between spikes and CSD is not quantitatively evaluated – substantial and minor changes should be demonstrated quantitatively. Claim: It is possible to alter the CSD pattern without substantially changing the firing rates by adjusting synaptic placement. One successful example is provided for this claim: putting all the excitatory inputs to layer four excitatory neurons onto their basal dendrites changes the contribution to CSD without changing the firing rate. Claim: it is possible to alter the firing rates without substantially changing the CSD pattern by adjusting synaptic weights. The example supporting this claim is not quantitatively verified and is not obvious that the change in the CSD pattern is minor. "LFP can reveal deficiencies in the model architecture not evident from firing rates. Models can be optimized for firing rates and CSD independently". Independent optimization is a stronger claim than what can be concluded from the results. The two-way dissociation between spiking and CSD is based on very few simple examples, and the dissociation is not perfect even then. The statistical metrics used to evaluate the change in the rate or CSD patterns are not adequate for such a strong conclusion.

Feedback can shape the sustained sinks and sources in superficial layers

Indeed, the final model simultaneously reproduces the evoked firing rate and CSD patterns in response to bright flash stimuli within animal variability, however, the metrics used cannot be directly interpreted. In essence, this important hypothesis is supported by a very indirect chain of results: experimental high-order firing rate data is fed into the model, which generated CSD/rate patterns, which are compared to the experimental CSD/rate patterns. In doing so the variability of the responses (CSD parameters of the superficial dipole in response to LM firing) provided by single trials and single animals is lost. The direct causal analysis would make this point more convincing.

Utility of CSD in constraining the model. The authors claim that CSD can be utilized to constrain the model. But even in this case, it serves as an after-the-fact confirmation of model performance by incorporating feedback from higher visual areas. Just using preexisting knowledge and experimental data to add feedback from HVA to the model and improving the CSD does not show the utility of CSD in tuning the model parameters. Especially since authors cut recurrent connections when investigating the contribution from external inputs to the CSD pattern it is unclear if feedback dipole cannot be mimicked by stronger excitatory/inhibitory synaptic input from the local circuit or slow delayed currents produced by active conductances. In other words, necessity and sufficiency cannot be demonstrated. Thus the approach chosen here could simply illustrate some ideas that might be further tested with proper analysis and experiments.

Many of the remarks above contained suggestions, few more technical ones are listed here.

An alternative to Wasserstein distance used between CSD patterns, is fitting a mixture model to sinks and sources and comparing the distribution of the parameters (location, time, half-width) of individual components in experimental data to that in the model. ICA decomposition, though now ideal, appears to be a good candidate. The notion of the canonical CSD pattern should rather be replaced by the statistical moments of the distributions of parameters of individual dipoles.

It would be more accurate to capture separately the magnitude of the rate increase and temporal aspects of the rate change (e.g. peak time, curvature of the rate increase curve) for each cell. Such measures, in addition, would replace clumsy aggregate firing rate population statistics computed (currently with KS-similarity) for different slices of time within the evoked response and would reflect spatio-temporal dynamics of the V1 circuit output.

KS-similarity could be replaced by more explicit measures capturing change in the distributions of the peak firing rate across populations. Temporal dynamics in the population can likewise be captured via characteristics of the distribution of the single neuron's peak latency. Statistical moments or direct characteristics of the PDF/CDF can be used for this quantification.

Overall both CSD and neural firing analysis marginalized by layer might not capture the key difference in the V1 circuit response. I believe that a multivariate analysis of the CSD and rate dynamics across layers that emphasizes relative changes of magnitude and latency of neural responses between the layers would provide a more principled measure. While firing rates might vary across experimental animals within each layer, between-layers relative changes in responses might be more consistent. Similar between-layer relative measures can be computed from the model data.

Feedback from higher visual areas. Experimental data provide a possibility for direct analysis of the origin of the superficial dipole: computing spike-CSD coherence (or spike-triggered average CSD) using spikes in higher-order visual areas and CSD in V1 allows for identifying which dipoles are consistent with the causal influence of the higher order visual cortex neural firing. Causal analysis of the firing rates in higher visual areas and CSD sink/source values on a single trial level would potentially allow for more precise conclusions. At any rate, even if correlational, this analysis would allow identifying which components of the CSD are correlated with the top-down inputs firing rate.

After adding feedback, the weights from the background node to feed-back targeted neurons are adjusted, but how is this justified? Please, clarify.

Diminished L4 sink in the final model. "The synaptic weights for thalamocortical connections were left unchanged from the original model" – and maybe that's why the early L4 sink is not prominent in the final model. It is striking that in most simulations the most prominent component in previously published and present evoked CSD profile, the L4 sink, is very weak in the simulation here. As suggested above, it appears that the model should reproduce the balance between various inputs' strengths to different layers manifested in the different CSD sink magnitudes. It would then be important to recapitulate this between-layer relative parameter of the input strength in the model (see comments on that above as well).

It is unclear why 90 minutes for 1 second of simulation presents such a barrier. When compared to experimental work this time seems realistic to provide much more statistically powerfull dataset using simulations. I would suggest increasing the number of trials of simulation.

Recommendations for improving the writing and presentation.

The authors compare the adjusted model to the original model but never mention what the original model was tuned to. "The synapses for recurrent connections were placed according to the following rules in the original model (Billeh et al. 2020)". As a reader, it would be helpful to have an idea of what the rules are based on. Is synaptic placement based on anatomy? Are synaptic weights fitted to evoked response to a particular visual stimulus?

Discussion should include a much more balanced interpretation of the results – it is merely an illustration of some ideas using the model, at the moment, rather than quantitative analysis demonstrating novel results on the composition of the CSD response, importance of the feedback, or recurrency in the actual experimental data. Likewise, the mismatch of the firing rate and CSD is simply illustrated by some examples, rather than revealing a novel result. In fact, the role of the interneurons in this mismatch is likely the key, but it was not discussed. I cannot imagine how a lack of realism in the interneuronal connectivity could be ignored (different PV morphological types, SOM, VIP) if one wants to recapitulate the circuit dynamics in response to the thalamic input. In other systems, for example, hippocampus, the mismatch between the CSD of theta and firing rates of different neuronal groups in entorhino-hippocampal circuits is much better understood and could be compared to here.

Discussion of other inputs not yet accounted for in the model that contribute to the experimental results is missing. What about other thalamic nuclei? Role of cortio-thalamic feedback?

It is clear that further improvements in the model realism will make it prohibitively complex. Authors postpone the future proper multiparametric optimization due to computational complexity. It would thus be useful to discuss the alternative approach – capturing parameters of the low dimensional dynamical system that would recapitulate the observed data or model

*Reviewer #3 (Recommendations for the authors):*

The authors present analysis of openly available Neuropixel data from the mouse primary visual cortex (V1), and several subcortical areas, provided by the Allen Institute. They seek to explain the detailed neuromechanistic underpinnings of flashed evoked responses in V1 using the previously established large scale V1 model by Billeh et al. 2020, also distributed by the Allen Institute. The authors establish a "canonical CSD pattern" occurring in V1 during the evoked response and work through a sequence of parameter manipulations needed to fit the model to the canonical CSD pattern, while simultaneously accounting for empirically observed firing rate activity in several populations of neurons. The main conclusions from the paper are that (1) feedback from higher order visual areas was necessary to account for both the early-latency (<100 ms) CSD and spiking features in the data, and (2) there is a two way disassociation between firing rates and CSD such that firing rates can be altered with minor effects on the CSD pattern by adjusting synaptic weights, and CSD can be altered with minor effects on firing rates by adjusting synaptic placement on the dendrites. The paper is well written with exciting new findings that are highly impactful to the field. It uses exceptional data and a phenomenally biophysically grounded and detailed large-scale model to make previously unattainable mechanistic interpretations of the data.

There are several directions that could be explored and clarified further that would increase the understanding and impact of the paper. Further, there is a history of earlier literature examining CSD/LFPs from visual evoked responses in VI in non-human primates that comes to similar conclusions on the necessity of "feedback" drive that should be discussed in comparison to the mouse results here.

1) While quantification of the activity of different cell types in different layers is necessarily limited in the empirical data, in the model it is not. The model contains more than 50,000 biophyscially detailed neuron models belonging to 17 different cell type classes: one inhibitory class (Htr3a) in layer 1, and four classes in each of the other layers (2/3, 4, 5, and 6) where one is excitatory and three are inhibitory (Pvalb, Sst, Htr3a) in each layer. In constraining the model to a limited number of features in the data (firing rates of limited cell populations and CSD), the model also makes many more novel and testable predictions about the activity of other cell types and synaptic connections (not fit too) that would be valuable for the readers to see.

For example, while the data is sparse in terms of knowledge of different inhibitory neuron types across populations, the model is not, and can provide testable predictions on the firing activity of the inhibitory neurons in each layer and across three different cell types Pvalb, Sst, Htr3a, along with Htr3a neurons in layer 1. In the final fit model as in Figure 7, it would be useful to detail model predictions about the firing rates in the 4 different interneuron types, and their strength of influence on the RS neurons. These predictions could then be further tested in follow up studies, and perhaps there is a prediction that was not used for fitting but that can be tested in the data set here?

Along this line, what potential mechanisms govern the animals or trials that present atypical CSD or spike rate evoked responses? The methods and results of this study are uniquely positioned to not only account for the circuit mechanisms of a canonical response, but also the cell and circuit mechanisms of a distribution visual flash evoked responses, particularly those categorized as non-canonical. How varied are the non-canonical responses? A supplemental figure showing the CSD from each of the animals, e.g. as shown for other features in Figure S4, would also be helpful to understand the full temporal spatial variability in the CSD. The present study has a distinctly powerful opportunity to make predictions regarding the underlying cell types, circuit elements, connectivity patterns, etc. that control for visual evoked response variability.

Related to this, there are a few important assumptions in the model that should be clarified further:

a. Are the LM to V1 connections known to not target L1 inhibitory neurons? What happens if they do in the simulations?

b. Do the "feedback" inputs in the model also contact the distal dendrites of the L6 RS cells? What about the apical dendrites of the L4 RS cells? What about the L1 cells? Is there any literature supporting the choice to connect the feedback to only a subset of the neurons in the supragranular layers or was it purely because it was necessary to reduce the magnitude of the L5/L6 dipole?

2) There are important details in the comparison made between the model neuron firing rates and empirically recording firing rates that need to be understood more fully. In the empirical recordings, cells are sorted into RS or FS populations, and RS are presumed excitatory with FS presumed inhibitory. The activity of the RS is separated by layer, but the FS is grouped across layers because of the sparsity of recordings. When comparing the model firing rates to the experimental firing rates, the excitatory and "non-Pvalb populations" were grouped together in each layer of the model to make up the RS cells in L2/3, L4, L5, and L6, while the Pvalb cells across all layers were grouped together to make up the FS cells of V1. This makes sense for the experimental data, however, the model has distinct non-Pvalb inhibitory populations including SST and Htr3a. Were these non-Pvalb inhibitory cells grouped with the FS cells or RS cells in the model analysis? Clarification is needed.

Line 557 states that to bring the firing rates of the neurons in the original model closer to the experimental data ["We first reduced the synaptic weights from all excitatory populations to the fast-spiking PV- neurons by 30% to bring their firing rates closer to the average firing rate in this population in the experiments. This resulted in increased firing rates in all other (RS) populations due to the reduced activity of the inhibitory Pvalb-neurons (Figure S6). Therefore, we further applied reductions in the synaptic weights from all excitatory neurons to RS neurons or increases in the synaptic weights from inhibitory neurons to the RS neurons to bring their firing rates closer to the experimental average firing rates." ] The "or" statement here is important, should it be "and"? Was it all inhibitory neurons or only the Pvalb-neurons?

A table with the final parameter values would be immensely helpful in interpreting the microcircuit details contributing to the final model fit in Figure 7.

3) The process of parameter tuning in models, particularly in large-scale models, is challenging. Throughout the results, the authors walk through a series of manipulations to the original model that are necessary to account for the empirical data. While I understand that not all parameters can be estimated, and there is a nice paragraph in the discussion on this starting on Line 981, it would be helpful to have a methods section that describes the overall strategy for parameter manipulations and rationale for the choice of the final parameters. Was it all hand tuning, or was there some estimation method used to fit to data? More specifically, there is a section called "Adjusting synaptic weights' that gives a high level description of a range of multiplication factors over which previous model parameters were adjusted to fit the current data. More details on how this range was determined, and if one value in this range or multiple values were ultimately used in the simulations, and why, is needed.

Details on how the new feedback was implemented in the model is described on Lines 682-690, but I didn't see any description of these inputs in the method section. The reference to Figure S3 seems to be a mistake. I suggest double checking all Figure references. It seems the "Adjusting synaptic weights" section only describes local interactions. Were there feedback inputs in the original Billeh model that were adjusted here? Or are they completely new, and if so, what were the starting weights and how were they chosen?

The modeling results here build from the prior work in Billeh et al. 2020. In this paper, the model is applied to visual flash evoked responses, while in Billeh et al. 2020 the model was tuned to orientation and direction tuning to drifting grating stimuli. It is exciting to see that the updated model reproduces consistent results for firing rates and rate based direction/orientation selectivity, as shown in Supplementary Figure S9. It is stated that for these simulations, the stimulation of the network was as defined in Billeh et al. 2020. One important piece of information to understand is why feedback is needed in the current simulations of flash evoked response, but isn't required for the drifting grading simulations? Would including feedback improve the results? A discussion and/or table summarizing the differences in the two models would be useful to understand what further manipulations may be necessary to accurately account for the LFP/CSD, in addition to firing rates, in Billeh et al. 2020 simulations.

4) One strategy the authors used to examine the importance of various neural mechanisms to the CSD, is to remove various model elements such as synaptic connections or intrinsic ionic currents and see how much the CSD changes. For example, there is a conclusion line 803 that [without inhibition that fact that] "the major sinks and sources are still present is an indication that the currents from excitatory input account for the majority of the sinks and sources observed in the experimental CSD." This could be due to the fact that in the fit model (Figure 7) the inhibitory synapses are very weak (are they?) and hence not playing a major role in the established model configuration. The conclusion that inhibition plays a small role is counterintuitive to the growing literature showing the importance of inhibition to LFP generation (e.g. Telenczuk et al. Scientific Reports 2017), and hence CSD, and further discussion and understanding of the role of the different types of inhibition in this model would be helpful.

A more useful analysis to quantify the impact of specific network features would be a parameter sensitivity analysis that shows the influence of changes in parameters to variance in the resultant CSD; this isn't feasible for all parameters but could perhaps be done for the parameters/network features concluded to account (or not account) for the main feature of the canonical CSD, e.g. as summarized beginning on line 827. Or at least some simulations that show what happens over a range of parameter values for the important parameters.

There is a reference in this paragraph to Figure 7E, which should be Figure 7F. The authors need to double check all Figure references.

5) A similar large scale biophysically detailed cortical column modeling paper (Reimann et al. Neuron 2013) is relevant to discuss further in light of the current results. In particular, that paper showed that "active currents" dominated the LFP, rather than synaptic currents. There is a statement line 890 that "In line with observations made by Reimann et al. (2013), we found that the somatic voltage- dependent membrane currents significantly shape the CSD signature (Figure 5H and 7B)". To my knowledge, Riemann considered the contribution of all active currents, including those in the dendrites, while in this paper they have only explored removing them in the soma. Those in the dendrites may have a greater impact, or maybe not (?). Can the authors show this? Further examination/discussion would be useful.

5) Of particular relevance to these results are earlier laminar recordings to flash evoked responses in VI in non-human primates that suggest the presence of "feedback" and discusses interaction between higher and lower order areas that may contribute to this "feedback": doi: 10.1016/0042-6989(94)90156-2 https://doi.org/10.1093/cercor/8.7.575. A discussion on the conclusions in this paper related to those is needed.

These papers, and many other papers, also show a decoupling between LFP/CSD and firing (MUA). Clarification should be made that such decoupling is not a new concept, but to my knowledge it has yet to be shown how it could be mechanistically implemented using a computational neural model, as the authors have nicely done. Nor has it been shown how simulated LFP/CSD can be altered with only minor effects on the firing rates, as shown here.

Also relevant, is a more recent study of flash evoked responses in humans and non-human primates doi: 10.1126/sciadv.abb0977, which appears to show consistent results to those reported here.

6) There is a paragraph in the methods on line 1644 that begins "This application of the WD to compare CSD patterns comes with certain considerations that are important to note…". I suggest moving this paragraph to the Results section.

---

## [Author Response]

[Editors’ note: the authors resubmitted a revised version of the paper for consideration. What follows is the authors’ response to the first round of review.]

Comments to the Authors:We are sorry to say that, after consultation with the reviewers, we have decided that this work will not be considered further for publication by Life. As you see there are a great number of technical concerns raised by the reviewers (in particular reviewer 2) which we feel, at present, preclude publication. However, we are happy to receive a resubmitted version of a strongly improved manuscript in which it becomes clear that you can convincingly address these technical concerns. In this we recommend writing an action plan of how you would address these concerns in advance.Reviewer #1 (Recommendations for the authors):While the dissection of CSD components is detailed and exhaustive, the analysis of unit firing response falls short. In previous works (like Senzai et al., 2018) a stereotypical timing of single unit firing to this type of stimuli is described in addition to the CSD responses. In the paper here can the author's model reproduce this timing of unit activation? In addition to reporting average firing rates, the authors should also describe and interpret the temporal dynamics of unit firing and relate it to their mechanistic description of CSD components in the last section.

The question of how to best characterize spike statistics is an important one, and we thank the reviewer for raising it. We acknowledge that there are relevant aspects of spike statistics that were not captured with the metrics we originally utilized. Reviewer 2 raised similar points, so per the suggestions of reviewers 1 and 2, we have supplemented our existing metrics with the following analyses:

Comparison of the moments of distributions of peak firing rates across cells in the evoked responseComparison of the moments of distributions of latencies to peak firing rate across cells.Comparing moments of the relative magnitude of firing rates across laminar populationsComparing moments of distributions of maximum curvature of firing rate

Summary of the results from (i):

For the first moment (mean) (Figure 6 —figure supplement 2A), the original model was outside the experimental variability for all populations except L5 RS. The final model was just outside of the experimental variability for the first moment of the peak firing rate of L2/3 RS (regular-spiking) cells, but inside for all other populations. The intermediate model was within experimental variability for all populations.

For the second moment (variance) (Figure 6 —figure supplement 2B), all model versions were within experimental variability for all populations.

For the third moment (skewness) (Figure 6 —figure supplement 2C), all three model versions were outside of the experimental variability for L2/3 RS cells, and the original model was outside the experimental variability for L5 RS and FS cells as well. For all other populations, all models were within experimental variability.

For the fourth moment (kurtosis) (Figure 6 —figure supplement 2D), all model versions were within experimental variability for all populations.

Summary of results from (ii):

All model versions were within experimental variability for all populations and all moments except the 4^th^ moment for FS cells, where all model versions were outside the experimental variability (Figure 6 —figure supplement 3)

Summary of results from (iii) and (iv):

For (iii) and (iv), all three model versions were within the experimental variability on all measures (Figure 6 —figure supplement 4 and Figure 6 —figure supplement 5).

Some of the analysis and simulations can be better motivated. For example, it is not clear, in the section starting in line 244, why it is important to quantify CSD variability and how that relates to the broader goals of the study. The manuscript in quite dense and in occasions is not easy to follow how certain results contribute to the main objective. It would help to state these explicitly at the outset and related each subsequent section to them.

The numbers we obtain when quantifying similarity (or distance) between the model and experimental targets cannot necessarily be easily interpreted on their own, so we quantify CSD (and spike) variability in experiments to use as references in our assessment of how similar the signals produced by the model are to those recorded in experiments. Calculating Wasserstein distances from CSD of individual mice to the PC 1 canonical CSD target, for example, gives us an idea of how big of a Wasserstein distance we can expect in modeled CSD that reproduces experimental observations. Calculating pairwise Wasserstein distances between animals provides a reference to use to judge what constitutes a large and a small change in CSD patterns from one model version to the next, which we did in Figure 5 —figure supplement 1. We added a short description of this motivation on lines 247-248 and 351-353.

When discussing the biophysical mechanisms of CSD generation a useful concept to introduce is that of dipolar moment. Components with a larger dipolar moment (more separation between sink and source) generate larger LFPs.

We added the following brief description of this mechanism in the context of adjusting the position of excitatory synapses onto the excitatory L6 cells on lines 702-704:

“Greater separation between a sink and a source in a dipole moment increases its magnitude, so restricting the range within which synapses can be placed should diminish the L5/L6 dipole’s dominance.”

Reviewer #2 (Recommendations for the authors):1. Spiking patterns alone, they argue, do not uniquely characterize a neural circuit, i.e. there are many models (or variations to the same model) that are capable of reproducing the spiking activity of data, so requiring the model to simultaneously reproduce the CSD would further restrict the model.

We do argue that requiring the model to match CSD in addition to spiking patterns would further constrain the model, that is correct; however, we want to stress that a model that reproduces both spiking activity and CSD is also unlikely to be uniquely characterized, even if it is more restricted than a model that only reproduces spiking activity. Gutenkunst et al., 2007, among others, have demonstrated that many different parametrizations can produce models that fit the data equally well in systems biology (so-called parameter “sloppiness”). This has also been shown for multi-compartment models in neuroscience; e.g. Hay et al., 2011, found that more than 500 parametrizations of their biophysically detailed layer 5 pyramidal neuron could perform the same function equally well.

2. The key novelty of this manuscript compared to a previous paper (Billeh et al. 2020) is adding CSD patterns as an additional constraint to more accurately infer the model parameters beyond its prior state. While it is true that adding constraints would undoubtedly be helpful, there is no guarantee that specific hyperparameters used, components of the model simplified or ignored and gross aggregated measures chosen for comparing the patterns in the data and produced by the model would allow identifiability of the model and use it to explain the experimental data.

We agree that identifiability of a true model, in the sense of a model where all parameters correspond to true experimental values, cannot be guaranteed. Nor can it be expected. Limitations on the data presently available as well as on computational resources necessitates simplifications that restrict the realism of the model. However, this is a feature of all modelling efforts in neuroscience. We would argue that despite these caveats, models, including the one presented here, can still give valuable insight into the mechanistic underpinnings of experimental observations.

3. Authors make no justification for many specific choices for a number of unknown high-dimensional parameters e.g. morphology of particular cell types (e.g. L4 pyramids have minimal basal dendrites, Scala et al. 2019, which has a strong effect on Figure 5 results), somato-dendritic distribution of synaptic inputs from different afferents, choice of interneurons (morphology, intrinsic properties, and connectivity) to include and how to wire them in recurrent and feedforward circuit, etc.

These choices, with the exception of the choice to use L4 pyramids for the results in Figure 5, were not made in this study but in Billeh et al., 2020 and Gouwens et al., 2018, on which this investigation builds. The work reported in those papers was extensive and meticulous, relying as much as possible on systematically obtained experimental data. For example, morphologies of particular cell types were taken from the Allen Cell Types Database, selecting such morphologies from all the cells for which high-quality models were available. The choices that are most relevant to the interpretation of our findings in this paper and their justification are relayed in the Methods section. The choice to use L4 pyramidal cells in Figure 5 was somewhat arbitrary. We have therefore carried out the same simulations and analysis on L2/3 and L5 excitatory cells for comparison and added plots of the results in the supplementary material (Figure 5 —figure supplement 3 and Figure 5 —figure supplement 4).

4. Given the nonlinear nature of the network dynamics driven by the thalamic input and its nontrivial dependence on very high-dimensional parameter space, I see no way to identify a true model in that space based on very coarse metrics of aggregated neural activity. Care for realism in the model and the use of real data provides a false impression that the outcome of the effort should be the identification of the actual unobserved parameters of the system and the identification of the model. I think that it is a false impression, that might be misleading the readers.

In the initial manuscript, we noted how the nonlinear nature of network dynamics limits the precision with which we can estimate the contributions of different pathways to the specific current sinks and sources, and hence, the parameters involved in their generation (see its lines 962-967 or lines 1042-1047 in the one presently submitted). Thus, this is a view we share with the reviewer. We have supplemented our existing metrics with other measures for comparing the neural activity of the model to experiments. These are described in further detail in points 6, 7, and 9 below.

However, even with these considerations, we would argue that our model can still shed light on which parameters are involved in various experimentally observed phenomena, and as more data becomes available in the future, the precision in our estimation of these parameters will improve.

5. The choice of metrics and dimensionality reduction approaches used to capture the group results on the CSD and firing rate is not suitable for validating the model or might provide poorly interpretable results. The authors note in the discussion, that there might be more suitable metrics and promise to work on it in the future. However, if the choice of metric used in the present paper makes interpretation questionable it puts in question the validity of all the results and conclusions in the manuscript, except for the actual data and model it is based on. Thus, I find it critical that the analysis methods used are chosen carefully and shortcomings discussed below are taken into account.

This is addressed in points 6, 7, and 9 below.

6. Inter-animal CSD alignment. It is obvious from the examples and prior work (Senzai et al. 2019) that variation of the location and scale of many CSD dipoles varies greatly. Any further analysis that relies on the CSD has to be thus rescaled and realigned, as Senzai et al. paper did (see their Figure 1). This would allow capturing the true variability due to physiological structure across all the data and not the variability related to the anatomical location of the electrode. I believe that adding this step would very strongly improve the remaining analysis (see more below).

The reviewer correctly points out that the CSD of each animal should be aligned and scaled to be comparable across animals and to ensure that our analysis does not merely reflect variability unrelated to the physiological differences between animals. In our initial manuscript, we did align and scale the CSD of all animals. However, in contrast to the approach in Senzai et al., 2019, we aligned the probes based on the depth of the electrode obtained from histology and provided in Allen Common Coordinate Framework (CCF) coordinates rather than landmarks in the data.

In this revised manuscript, we have implemented an alignment of the CSD based on landmarks in addition to our original alignment based on histology, and we supply the results from a quantitative comparison of the model CSD to the experimental CSD after landmark alignment (Figure 6 —figure supplement 7). Our conclusions regarding which model versions reproduce the experimental CSD patterns and which do not are unchanged by using the landmark alignment approach.

7. Canonical CSD pattern. The choice of the first principal component of trial averaged CSDs as the canonical CSD pattern is not at all obvious and has potential ramifications for the remaining analysis that it is based on. The first principal component is capturing the direction of inter-animal variability in the CSD pattern time-depth space affected by the angle of the probe, alignment of the recording sites array to cortical layers, exact retinotopic location, variation of brain states across individual recording sessions etc. This first eigenvector of the covariance matrix does not necessarily capture a common underlying pattern. In fact, mathematically it is not simply a weighted average but depends on the structure of the remaining unexplained variance (ca 50%). The components of the evoked response that strongly covary would contribute more to the first PC, but those that are uncorrelated or partially independent would contribute in an unpredictable way.

The reviewer is correct that, mathematically, the first principal component is strictly speaking not a weighted average. We have updated the relevant parts of the manuscript such that it reads “weighted contributions” rather than “weighted average”.

The question of how probe alignment could affect this analysis is addressed in point 6 above.

We have added CSD plots for all animals (Figure 2 —figure supplement 1), a plot of the cumulative explained variance of all principal components (Figure 2 —figure supplement 2A), as well CSD plots for the 10 first principal components that cumulatively explain 90 % of the variance (Figure 2 —figure supplement 2B) in the supplementary material. This way, an assessment of whether the first principal component captures a common underlying CSD pattern is available to the reader.

We have also added the analysis where we used a plain average, rather than the first principal component, of trial averaged CSD across animals as the target in our quantitative analysis of model performance. We did this both where the animal CSDs had been aligned to histology, as it was originally, and where they had been aligned to landmarks in the data. In both cases, the model reproduced the experimentally observed CSD pattern and supported our conclusions from the analysis where we used the first principal component of the histology aligned CSD as the target (Figure 6 —figure supplement 7C and Figure 6 —figure supplement 9).

8. The Wasserstein distance used as a measure of distance between two CSD patterns is sensitive to misalignment of the CSD patterns (see above). The WD statistics cannot differentiate the contribution of the single sink and source components as they are aggregated into a single compound measure to which both anatomical variability contributes potentially more than the physiological variability. At the same time, the key elements of the paper are critically linked to this measure and its ability to capture the similarity between the experiment and the model. An alternative to Wasserstein distance used between CSD patterns, is fitting a mixture model to sinks and sources and comparing the distribution of the parameters (location, time, half-width) of individual components in experimental data to that in the model. ICA decomposition, though now ideal, appears to be a good candidate. The notion of the canonical CSD pattern should rather be replaced by the statistical moments of the distributions of parameters of individual dipoles.

It is true that ICA has been successfully applied to distinguish contributions from distinct pathways to recorded LFP in several cases (e. g. Makarov et al., 2010, Makarov et al., 2011, Fernandez-Ruiz et al., 2012, Fernandez-Ruiz et al., 2017). However, in these examples, ICA was applied to LFP recorded in subcortical structures such as hippocampus, and the specific conditions that allow for its successful application in those structures may not hold for other areas of the brain, such as V1. In Makarov et al., 2011, the factors that reduce ICA’s performance are listed and include high synchronicity and a strong imbalance in the relative power of the LFP generated by different biophysical processes. If the various generators of LFP are synchronously activated or their contributions are imbalanced, significant cross-contamination of the LFP attributed to them with ICA may occur. In our data, there is substantial temporal overlap between the biophysical processes during evoked flash response in V1, which would likely produce such cross-contamination. Whether or not there are also strong imbalances in relative power of LFP generated by different biophysical processes is not known for sure and would constitute another potential source of cross-contamination when applying ICA.

Furthermore, the underlying assumption of independence of LFP generators is unlikely to hold true during flash-evoked responses in V1. LGN contributes to all layers in mouse V1, even if L4 is its primary target. This means that there may be significant contributions from LGN to several different sinks in V1. In fact, our own analysis suggests that LGN, together with recurrent connections, is an important contributor to both the L4 and the L5/L6 sink. Thus, the biophysical processes generating the L4 sink and the L5/L6 sink are not independent. Additionally, even if the direct contributions of the LGN input to the L5/L6 sink should be negligible, the recurrent contribution to the L5/L6 sink is dependent on prior LGN input to L4, so the recurrent generation of the L5/L6 sink would in any case not be a pathway that is independent of the LGN input pathway. The same is likely to be true for the input from HVAs such as LM, as they receive input from V1 (Harris et al., 2019, Siegle et al., 2020). Therefore, one cannot expect ICA to successfully separate the different sink/source dipoles into different components.

Note also that Glabska et al., 2014 tested the application of ICA on LFP generated generated by the Traub model (Traub et al., 2005) of somatosensory cortex, and found that no single component could recover the activity of a single population. They could only recover the population activity through combining components, and, even then, only two populations could reliably be recovered, at most three. They also point out that determining which components to combine depended on knowledge of the ground truth population activity in the model, something one does not have access to in the experiments.

We still tested the application of ICA on our experimental and model data. Example results from two animals and the final model are shown in Author response image 1. In accordance with expectations from Makarov et al. 2011, there appears to be significant cross-contamination between the components both for the animals and the model, as similar patterns recur in several components (see e. g. IC 1 and 2 in Author response image 1), and one component picks up most of the variance (see e. g. IC 2 for the second animal and IC 3 for the final model). It is also not clear which component should be considered to correspond to which biophysical process, nor which components should be compared across animals and between animals and the model. Utilizing all trials instead of trial averaged data or a different number of components did not significantly change the results.

**Author response image 1. sa2fig1:** ICA decomposition of experimental CSD and simulated CSD. (A) Two examples of applying ICA on trial averaged CSD of animals. Four resulting independent components are displayed from left to right, and the trial averaged CSD of the example animal is shown in the rightmost plot. (B) Four independent components resulting from applying ICA to trial averaged CSD from the simulation on the final model version.

Lastly, we want to note that we are aware that in Senzai, Fernandez-Ruiz, and Buzsáki, 2019, ICA was successfully applied to decompose LFP recorded in V1. However, in that study, ICA was applied after filtering in the γ band (30-100 Hz), as its purpose was to identify layer-specific γ oscillations. Filtering in the γ band first is not appropriate in our study, since we are equally interested in signals outside the γ range. Thus, even though ICA was applicable to V1 data in Senzai, Fernandez-Ruiz, and Buzsáki, 2019, that does not imply a suitable decomposition of LFP/CSD in our study.

The Wasserstein distance metric is not without limitations, which we note in the manuscript. However, it does not require any human characterization of which sinks and sources should be paired and compared across animals and between animals and the model. We wanted to employ a metric that could compare whole CSD patterns; that is, the relative strength, shape, timing and position of sinks and sources without such, often subjective, pairing. Thus, a compound measure like the Wasserstein distance was in our view the most appropriate.

9a. KS-similarity. This statistic captures coarse (max of CDF difference) differences of the CDFs and is used to quantify the variability of firing rate magnitude across different populations. As a distance between two distributions, KS distance is not capturing the shape of the distributions and it's not quantifying where they are different. One can see that KS-similarity score is close to 0.75 for all populations in the dataset – the specificity of this measure is low. Importantly, this distribution similarity measure is not sensitive to the nature of change (direction), affected by the emergence of skew or modes in the distributions etc. Critically, the application of the measure to distinct layers doesn't capture joint spatial patterns of the firing rate across layers, which could get it closer to the CSD similarity measure. As the direction of the rate change is not distinguished, very different spatial patterns of firing rate change would be associated with similar KS-similarity values.b. Pearson correlation is used to quantify the similarity in the average temporal profile of firing rates of layer-specific populations in experimental and model data. First, In contrast to the KS-similarity measures that strive to capture the distribution of the population firing rates, but ignores temporal dynamics, this measure ignores the variability of neural firing rates and their temporal dynamics. Changes in the shape of the distribution of firing rates will translate into unpredictable changes of the mean firing rate. Variable temporal profiles of neurons would further contribute to the poor interpretability of the mean firing rate evoked response. Given the generally log-normal distribution of the firing rates in many cortical circuits the mean is poorly describing the distribution. Overall, I can't see how mean rate response could be used as an interpretable statistics characterizing complex spatio-temporal dynamics across populations of neurons. A huge spread of confidence interval in all figures showing mean population PSTH is a testament to this point. Second, as the mean firing rate statistics is unlikely normally distributed, given the nonstationary changes of the firing rate, and might not be even unimodally distributed in the 0-100ms time window, using Pearson correlation coefficient is not ideal as it best measures the linear relationship between two normally distributed random variables. It is deviations from a bivariate normal distribution that would influence the value of correlation coefficient introducing a bias. It is hard to interpret what values ranging between 0.3 to 0.6 reflect if anything. Multiple spectral components of the evoked response would contribute in an unaccountable way to the measure further complicating the interpretation.c. KS-similarity could be replaced by more explicit measures capturing change in the distributions of the peak firing rate across populations. Temporal dynamics in the population can likewise be captured via characteristics of the distribution of the single neuron's peak latency. Statistical moments or direct characteristics of the PDF/CDF can be used for this quantification.d. It would be more accurate to capture separately the magnitude of the rate increase and temporal aspects of the rate change (e.g. peak time, curvature of the rate increase curve) for each cell. Such measures, in addition, would replace clumsy aggregate firing rate population statistics computed (currently with KS-similarity) for different slices of time within the evoked response and would reflect spatio-temporal dynamics of the V1 circuit output.e. Overall both CSD and neural firing analysis marginalized by layer might not capture the key difference in the V1 circuit response. I believe that a multivariate analysis of the CSD and rate dynamics across layers that emphasizes relative changes of magnitude and latency of neural responses between the layers would provide a more principled measure. While firing rates might vary across experimental animals within each layer, between-layers relative changes in responses might be more consistent. Similar between-layer relative measures can be computed from the model data.

The question of how to best characterize spike statistics is an important one, and we thank the reviewer for raising it. We acknowledge that there are relevant aspects of spike statistics that were not captured with the metrics we originally utilized. Reviewer 1 raised similar points, so per the suggestions of reviewers 1 and 2, we have supplemented our existing metrics with the following analyses:

Comparison of the moments of distributions of peak firing rates across cells in the evoked responseComparison of the moments of distributions of latencies to peak firing rate across cells.Comparing moments of the relative magnitude of firing rates across laminar populationsComparing moments of distributions of maximum curvature of firing rate

Summary of the results from (i):

For the first moment (mean) (Figure 6 —figure supplement 2A), the original model was outside the experimental variability for all populations except L5 RS. The final model was just outside of the experimental variability for the first moment of the peak firing rate of L2/3 RS (regular-spiking) cells, but inside for all other populations. The intermediate model was within experimental variability for all populations.

For the second moment (variance) (Figure 6 —figure supplement 2B), all model versions were within experimental variability for all populations.

For the third moment (skewness) (Figure 6 —figure supplement 2C), all three model versions were outside of the experimental variability for L2/3 RS cells, and the original model was just outside the experimental variability for L5 RS and FS cells as well. For all other populations, all models were within experimental variability.

For the fourth moment (kurtosis) (Figure 6 —figure supplement 2D), all model versions were within experimental variability for all populations.

Summary of results from (ii):

All model versions were within experimental variability for all populations and all moments except the 4^th^ moment for FS cells, where all model versions were outside the experimental variability (Figure 6 —figure supplement 3)

Summary of results from (iii) and (iv):

For (iii) and (iv), all three model versions were within the experimental variability on all measures (Figure 6 —figure supplement 4 and Figure 6 —figure supplement 5).

10. A small number of simulated trials yields very noisy (high variance) average CSD and PSTH. Given the importance for the interpretation of the results presented in the paper and the stochastic nature of the simulated data, it would be valuable to run longer simulations (more trials) to reach reliable low-variance estimates of the CSD/PSTH.

Since we use experimentally recorded spike trains as input to the model, the number of trials we can run in a simulation is limited by the number of experimentally recorded spike trains we have in the structures providing input to primary visual cortex. If we increase the number of trials, we have fewer spike trains to use to create unique inputs in each trial. And the fewer spike trains we use to create input for each trial, the more variable the input in each trial will be. Thus, increasing the number of trials will not necessarily reduce the variability in the simulations. We have experimented with different trial numbers and found that our current number gives a reasonable trade-off between the variability due to limited spike trains and variability due to a limited number of trials.

11. Analysis of the biophysical origin of the canonical CSD using the model is an interesting and worthy line of investigation. However, from the approach used in this paper that compares CSD due to one pathway to that generated by the full model, it is difficult to make conclusions about the exclusive contribution of that pathway, since this system with many recurrent connections is nonlinear. Figure 7 only shows single examples and relies on very specific, and somewhat arbitrary parameters that are critical for the outcome. Somato-dendritic distribution and density of synapses between a particular input and target neuron, the morphology of that neuron, and the strength of input all make a large impact on the resulting CSD. Most statements are based on a single example (i.e. specific parameters choice) and eye-balling, with no quantification provided. While it could provide some qualitative intuition for a plausible generative model of the CSD of the evoked response in V1, the large number of unknown parameters that were specifically chosen in the model makes it impossible to generalize the results of these single examples to the experimental data.

We agree that it’s difficult to make conclusions about the exclusive contributions of specific pathways due to the nonlinear effects of recurrent connections (see lines 1042-1047). Indeed, this difficulty was among the reasons why we limited our analysis to one that could provide a qualitative intuition for a plausible generative model of the evoked CSD response in V1 in the first place. Attempting to give precise and generalizable numbers for the contributions of specific pathways lies beyond what is possible with the model at this stage. The abovementioned limitation, as well as other simplifications of the model (e. g., that only the feedback from LM was modeled, while V1 also receives feedback from other higher visual areas), and limitations in the data presently available prohibits exact enumeration of the exclusive contribution from specific pathways to specific sinks and sources. Intuition for a plausible generative model of CSD is, in our view, a more appropriate representation of what can be interpreted from our results at this time.

12. The two-way dissociation between spikes and CSD is not quantitatively evaluated – substantial and minor changes should be demonstrated quantitatively. Claim: It is possible to alter the CSD pattern without substantially changing the firing rates by adjusting synaptic placement. One successful example is provided for this claim: putting all the excitatory inputs to layer four excitatory neurons onto their basal dendrites changes the contribution to CSD without changing the firing rate. Claim: it is possible to alter the firing rates without substantially changing the CSD pattern by adjusting synaptic weights. The example supporting this claim is not quantitatively verified and is not obvious that the change in the CSD pattern is minor. "LFP can reveal deficiencies in the model architecture not evident from firing rates. Models can be optimized for firing rates and CSD independently". Independent optimization is a stronger claim than what can be concluded from the results. The two-way dissociation between spiking and CSD is based on very few simple examples, and the dissociation is not perfect even then. The statistical metrics used to evaluate the change in the rate or CSD patterns are not adequate for such a strong conclusion.

We have now added quantitative evaluations of both the change in spiking with changes in synaptic placement alone and changes in CSD with changes in synaptic weights (Figure 5 – —figure supplement 2 and Figure 5 —figure supplement 1, respectively), and these evaluations support our original conclusions. The added quantification strengthens our analysis, and we thank the reviewer for making this suggestion. Furthermore, we have provided more examples of the effects of manipulating synaptic placement in Figure 5 —figure supplement 3 and Figure 5 —figure supplement 4. We want to note that our manuscript did not state that our results demonstrated a perfect dissociation and that our comment that models can be optimized for firing rates and CSD independently was preceded by the qualifier “to a certain extent”. We do not hold the view that CSD and spikes can be completely dissociated and be utilized entirely independently in optimization, as this standpoint can neither be supported by our own data and analysis nor previous work on this topic.

13. Feedback can shape the sustained sinks and sources in superficial layersIndeed, the final model simultaneously reproduces the evoked firing rate and CSD patterns in response to bright flash stimuli within animal variability, however, the metrics used cannot be directly interpreted. In essence, this important hypothesis is supported by a very indirect chain of results: experimental high-order firing rate data is fed into the model, which generated CSD/rate patterns, which are compared to the experimental CSD/rate patterns. In doing so the variability of the responses (CSD parameters of the superficial dipole in response to LM firing) provided by single trials and single animals is lost. The direct causal analysis would make this point more convincing.

This is an interesting idea, and we will consider it for future extensions of our work.

14.Utility of CSD in constraining the model. The authors claim that CSD can be utilized to constrain the model. But even in this case, it serves as an after-the-fact confirmation of model performance by incorporating feedback from higher visual areas. Just using preexisting knowledge and experimental data to add feedback from HVA to the model and improving the CSD does not show the utility of CSD in tuning the model parameters. Especially since authors cut recurrent connections when investigating the contribution from external inputs to the CSD pattern it is unclear if feedback dipole cannot be mimicked by stronger excitatory/inhibitory synaptic input from the local circuit or slow delayed currents produced by active conductances. In other words, necessity and sufficiency cannot be demonstrated. Thus the approach chosen here could simply illustrate some ideas that might be further tested with proper analysis and experiments.

We cannot, at this point, definitively rule out that the feedback from HVAs does not contribute to the late dipole in upper layers and that the origin of this dipole is predominantly local (stemming from activity driven by recurrent connections within V1). However, a few observations make this less plausible:

The late dipole is roughly 20 ms delayed relative to the onset of the initial response (which occurs about 40 ms after stimulus presentation). That’s a substantially longer delay than most local circuit mechanisms can explain. For example, the conduction delay of monosynaptic connections between granular and supragranular processes would not be longer than a few ms, and we can observe transient sinks in supragranular layers already at about 45 ms for many animals (Figure 2 —figure supplement 1), which we believe is more likely to reflect the local input from L4 to L2/3.The sustained L4 source could in principle originate in input from inhibitory populations. However, the firing rate of FS cells is decreasing in the time window after 60 ms (Figure 3A), which does not support the possibility of strong inhibition in this time window.All attempts to mimic the late dipole with changes in the excitatory/inhibitory synaptic input were unsuccessful, whereas providing feedback inputs from HVA readily helped to produce the late dipole resembling the one seen in experiments.

Of course, this does not mean that local circuit explanations of the late dipole are completely ruled out, but they do make them less parsimonious than the explanation by feedback from HVAs.

15. After adding feedback, the weights from the background node to feed-back targeted neurons are adjusted, but how is this justified? Please, clarify.

In the original model, the input from the background node represented the influence of the rest of the brain on V1, which included the influence from higher visual areas such as LM. This means that some of the feedback influence from LM on V1 should be present (though coarsely represented) in the input from the background node. When the influence of input from LM to feedback targeted cells is modelled on its own, the influence of the background node must be updated accordingly. This justification has been incorporated in the manuscript on lines 1703-1708.

16. Diminished L4 sink in the final model. "The synaptic weights for thalamocortical connections were left unchanged from the original model" – and maybe that's why the early L4 sink is not prominent in the final model. It is striking that in most simulations the most prominent component in previously published and present evoked CSD profile, the L4 sink, is very weak in the simulation here. As suggested above, it appears that the model should reproduce the balance between various inputs' strengths to different layers manifested in the different CSD sink magnitudes. It would then be important to recapitulate this between-layer relative parameter of the input strength in the model (see comments on that above as well).It is unclear why 90 minutes for 1 second of simulation presents such a barrier. When compared to experimental work this time seems realistic to provide much more statistically powerfull dataset using simulations. I would suggest increasing the number of trials of simulation.

The reviewer is right that the L4 sink appears too weak relative to the other sinks and sources when compared to what’s typically observed in our own and previously published experimental data. However, we would like to point out that there are several examples in the experiments we analyzed where the L4 sink is weak compared to the other sinks and sources (see session ids 816200189, 778240327, 799864342, and 794812542 in Figure 2 —figure supplement 1), so the model isn’t necessarily producing a CSD pattern that’s not biologically plausible, even if it is less stereotypical. Nonetheless, we intend to explore ways to make the L4 sink stronger relative to other sinks and sources in future work.

Regarding the relative strengths of different CSD sink magnitudes: this is already a part of the Wasserstein distance metric. For example, if the relative strengths of the sinks and sources in infragranular layers are too strong compared to sinks and sources in the granular and supragranular layers (as was the case in the original model), this is reflected in a greater Wasserstein distance (Figure 4B and 4D). The Wasserstein distance also allows us to explore how the sinks of one pattern are moved to match the sinks of another pattern (illustrated in Figure 2C) without any human characterization of what constitutes a L4 sink, a L2/3 sink, or a L5/L6 sink. This means that we already have a metric that allows us to assess the relative strengths of sinks in each layer.

The question of the number of trials in a simulation is addressed in point 10 above.

17. The authors compare the adjusted model to the original model but never mention what the original model was tuned to. "The synapses for recurrent connections were placed according to the following rules in the original model (Billeh et al. 2020)". As a reader, it would be helpful to have an idea of what the rules are based on. Is synaptic placement based on anatomy? Are synaptic weights fitted to evoked response to a particular visual stimulus?

A description of this has been added to the “Dendritic targeting” (on lines 1650-1651) and “Adjusting synaptic weights” (on lines 1695-1699) sections in Methods.

On lines 1650-1651:

“The rules for placement of synapses in Billeh et al. 2020 were based on reviews of literature on anatomy.”

On lines 1695-1699:

“In the original model, the synaptic weights of thalamocortical connections were based on experimental recordings of synaptic current, while the synaptic weights for recurrent connections were initially set to estimates from literature, then optimized to a drifting gratings stimulus until the model reproduced experimental values of orientation and direction selectivity.”

18a. Discussion should include a much more balanced interpretation of the results – it is merely an illustration of some ideas using the model, at the moment, rather than quantitative analysis demonstrating novel results on the composition of the CSD response, importance of the feedback, or recurrency in the actual experimental data.b. Likewise, the mismatch of the firing rate and CSD is simply illustrated by some examples, rather than revealing a novel result. In fact, the role of the interneurons in this mismatch is likely the key, but it was not discussed. I cannot imagine how a lack of realism in the interneuronal connectivity could be ignored (different PV morphological types, SOM, VIP) if one wants to recapitulate the circuit dynamics in response to the thalamic input. In other systems, for example, hippocampus, the mismatch between the CSD of theta and firing rates of different neuronal groups in entorhino-hippocampal circuits is much better understood and could be compared to here.

a) With the quantitative analysis added to the manuscript (see points 6, 7, 9, and 12 above), we consider the current interpretation of our results to be fair.

b) It is not clear why lack of realism is mentioned here. Our model includes distinct and experimentally constrained connectivity patterns for PV, SOM, and VIP neurons, parameterized individually for each layer based on experimental data. This realistic interneuronal connectivity was implemented in the model in Billeh et al., 2020, which we are using here as a starting point, as was different PV, SOM, and VIP morphological types.

19. Discussion of other inputs not yet accounted for in the model that contribute to the experimental results is missing. What about other thalamic nuclei? Role of cortio-thalamic feedback?

The effects of cortico-thalamic feedback should, at least partly, be present in the LGN input to the model, as we are using experimentally recorded spike trains as input. The role of other thalamic nuclei is beyond the scope of our study but would be an interesting avenue for future work.

20. It is clear that further improvements in the model realism will make it prohibitively complex. Authors postpone the future proper multiparametric optimization due to computational complexity. It would thus be useful to discuss the alternative approach – capturing parameters of the low dimensional dynamical system that would recapitulate the observed data or model.

We and other members of our research groups are working on approaches that can produce low dimensional dynamical systems that recapitulate the observed data or model, and the findings of these projects will be presented in future papers.

Reviewer #3 (Recommendations for the authors):1. While quantification of the activity of different cell types in different layers is necessarily limited in the empirical data, in the model it is not. The model contains more than 50,000 biophyscially detailed neuron models belonging to 17 different cell type classes: one inhibitory class (Htr3a) in layer 1, and four classes in each of the other layers (2/3, 4, 5, and 6) where one is excitatory and three are inhibitory (Pvalb, Sst, Htr3a) in each layer. In constraining the model to a limited number of features in the data (firing rates of limited cell populations and CSD), the model also makes many more novel and testable predictions about the activity of other cell types and synaptic connections (not fit too) that would be valuable for the readers to see.For example, while the data is sparse in terms of knowledge of different inhibitory neuron types across populations, the model is not, and can provide testable predictions on the firing activity of the inhibitory neurons in each layer and across three different cell types Pvalb, Sst, Htr3a, along with Htr3a neurons in layer 1. In the final fit model as in Figure 7, it would be useful to detail model predictions about the firing rates in the 4 different interneuron types, and their strength of influence on the RS neurons. These predictions could then be further tested in follow up studies, and perhaps there is a prediction that was not used for fitting but that can be tested in the data set here?Along this line, what potential mechanisms govern the animals or trials that present atypical CSD or spike rate evoked responses? The methods and results of this study are uniquely positioned to not only account for the circuit mechanisms of a canonical response, but also the cell and circuit mechanisms of a distribution visual flash evoked responses, particularly those categorized as non-canonical. How varied are the non-canonical responses? A supplemental figure showing the CSD from each of the animals, e.g. as shown for other features in Figure S4 (in the original submission, Figure 2 —figure supplement 3 in the current submission), would also be helpful to understand the full temporal spatial variability in the CSD. The present study has a distinctly powerful opportunity to make predictions regarding the underlying cell types, circuit elements, connectivity patterns, etc. that control for visual evoked response variability.

This is an interesting suggestion, and one we can pursue in future explorations of the model. Done thoroughly, this could create enough material to warrant another article on its own.

We have included a supplemental figure displaying the CSD from each animal (Figure 2 —figure supplement 1) to give the reader a better idea of the variability in CSD responses.

2. Related to this, there are a few important assumptions in the model that should be clarified further:a. Are the LM to V1 connections known to not target L1 inhibitory neurons? What happens if they do in the simulations?b. Do the "feedback" inputs in the model also contact the distal dendrites of the L6 RS cells? What about the apical dendrites of the L4 RS cells? What about the L1 cells? Is there any literature supporting the choice to connect the feedback to only a subset of the neurons in the supragranular layers or was it purely because it was necessary to reduce the magnitude of the L5/L6 dipole?

a) The literature does not suggest that LM to L1 connections are known not to target L1 inhibitory neurons. However, to our knowledge, only apical dendrites of neurons whose somata reside in L2/3 and L5 are mentioned as targets of connections that terminate in L1. Still, we have run simulations that included connections between LM and L1 inhibitory neurons. We have added plots that display the results from these simulations in Appendix 1 – Figure 1.

b): No, the feedback does not target L4, L6, or L1 cells. We connected the feedback only to L2/3 and L5 cells because those were the only cells that were mentioned as targets in the literature. We cannot rule out that LM targets cells in the other layers as well, but since the literature does not indicate these layers as important targets for LM afferents, we chose to leave them out for now.

3. There are important details in the comparison made between the model neuron firing rates and empirically recording firing rates that need to be understood more fully. In the empirical recordings, cells are sorted into RS or FS populations, and RS are presumed excitatory with FS presumed inhibitory. The activity of the RS is separated by layer, but the FS is grouped across layers because of the sparsity of recordings. When comparing the model firing rates to the experimental firing rates, the excitatory and "non-Pvalb populations" were grouped together in each layer of the model to make up the RS cells in L2/3, L4, L5, and L6, while the Pvalb cells across all layers were grouped together to make up the FS cells of V1. This makes sense for the experimental data, however, the model has distinct non-Pvalb inhibitory populations including SST and Htr3a. Were these non-Pvalb inhibitory cells grouped with the FS cells or RS cells in the model analysis? Clarification is needed.

The non-PV inhibitory cells were grouped together with excitatory cells to make up the RS populations in the model analysis, which corresponds to the current understanding of what the experimentally observed RS and FS populations are: FS are mostly PV inhibitory cells, whereas RS are the excitatory AND non-PV inhibitory cells. This is explained on lines 340-342 and 1572-1575 but we acknowledge that this could have been made clearer.

4. Line 557 states that to bring the firing rates of the neurons in the original model closer to the experimental data ["We first reduced the synaptic weights from all excitatory populations to the fast-spiking PV- neurons by 30% to bring their firing rates closer to the average firing rate in this population in the experiments. This resulted in increased firing rates in all other (RS) populations due to the reduced activity of the inhibitory Pvalb-neurons (Figure S6 in the original submission, figure 4 —figure supplement 1 in the current submission). Therefore, we further applied reductions in the synaptic weights from all excitatory neurons to RS neurons or increases in the synaptic weights from inhibitory neurons to the RS neurons to bring their firing rates closer to the experimental average firing rates."]a. The "or" statement here is important, should it be "and"?b. Was it all inhibitory neurons or only the Pvalb-neurons?

a): It should be “and”. The relevant part of the manuscript has been corrected.

b): It was only the Pvalb-neurons.

5. The process of parameter tuning in models, particularly in large-scale models, is challenging. Throughout the results, the authors walk through a series of manipulations to the original model that are necessary to account for the empirical data. While I understand that not all parameters can be estimated, and there is a nice paragraph in the discussion on this starting on Line 981, it would be helpful to have a methods section that describes the overall strategy for parameter manipulations and rationale for the choice of the final parameters.a. Was it all hand tuning, or was there some estimation method used to fit to data?b. More specifically, there is a section called "Adjusting synaptic weights' that gives a high level description of a range of multiplication factors over which previous model parameters were adjusted to fit the current data. More details on how this range was determined, and if one value in this range or multiple values were ultimately used in the simulations, and why, is needed.

a) It was all hand tuning. We postpone automatic estimation to future projects because the computational cost of running the number of simulations required for such methods is currently prohibitively high with this model.

b) The ranges were set after experimenting with different ranges to find the ones that would allow the model to reproduce the experimental observations. Only a single value within each range was used in the final model. We have added this information to the relevant part in the Methods section of the manuscript.

6a. Details on how the new feedback was implemented in the model is described on Lines 682-690, but I didn't see any description of these inputs in the method section.b. It seems the "Adjusting synaptic weights" section only describes local interactions. Were there feedback inputs in the original Billeh model that were adjusted here? Or are they completely new, and if so, what were the starting weights and how were they chosen?c. The modeling results here build from the prior work in Billeh et al. 2020. In this paper, the model is applied to visual flash evoked responses, while in Billeh et al. 2020 the model was tuned to orientation and direction tuning to drifting grating stimuli. It is exciting to see that the updated model reproduces consistent results for firing rates and rate based direction/orientation selectivity, as shown in Supplementary Figure S9 (in the original submission, Figure 6 —figure supplement 6 in the current submission). It is stated that for these simulations, the stimulation of the network was as defined in Billeh et al. 2020. One important piece of information to understand is why feedback is needed in the current simulations of flash evoked response, but isn't required for the drifting grading simulations? Would including feedback improve the results? A discussion and/or table summarizing the differences in the two models would be useful to understand what further manipulations may be necessary to accurately account for the LFP/CSD, in addition to firing rates, in Billeh et al. 2020 simulations.

a) A description of the feedback input can be found under “Input from lateromedial area (LM)” under “Visual stimulus” in the Method section.

b) There was no feedback input to the original Billeh et al. 2020 model. In the revised version of the model that we report in this manuscript, the starting weights were set equal to the weights from the background node to the feedback targeted population. This was done because the input from the background node was a representation of the influence from the rest of the brain before the feedback input from LM had been added, which means that in the original model background represented the influence from HVAs (in addition to influences from all other areas of the brain), and could therefore be seen as a simple, first approximation of the weights from LM.

c) Feedback was included in the drifting gratings simulation (Figure S9 in the original submission, Figure 6 —figure supplement 6 in the current submission) too. Though the simplicity of the connectivity and input from LM in this model meant that we did not expect the direction and orientation selectivity of the model improve, we included it to make this simulation more similar to the final model simulation of the evoked flash response, so that we could test if any of the previous features of the model had been “broken” by the inclusion of feedback.

7. One strategy the authors used to examine the importance of various neural mechanisms to the CSD, is to remove various model elements such as synaptic connections or intrinsic ionic currents and see how much the CSD changes. For example, there is a conclusion line 803 that [without inhibition that fact that] "the major sinks and sources are still present is an indication that the currents from excitatory input account for the majority of the sinks and sources observed in the experimental CSD."a. This could be due to the fact that in the fit model (Figure 7) the inhibitory synapses are very weak (are they?) and hence not playing a major role in the established model configuration.b. The conclusion that inhibition plays a small role is counterintuitive to the growing literature showing the importance of inhibition to LFP generation (e.g. Telenczuk et al. Scientific Reports 2017), and hence CSD, and further discussion and understanding of the role of the different types of inhibition in this model would be helpful.

a) We do not have reason to believe that the inhibitory synapses are weak relative to excitatory synapses in the model, as the model reproduces the experimentally observed firing rates of both fast-spiking PV-cells and regular-spiking cells. If the inhibitory synapses were too weak, one would expect the firing rate in excitatory (RS) populations to be significantly higher than the experimental firing rates in these populations. It is possible that both the excitation and inhibition are weak, which could produce firing rates compatible with experimental observation, but that should in any case not result in a disproportionate contribution to the CSD from the excitatory populations, since both excitatory and inhibitory synapses are weak in this scenario.

b) This is an important point. The following discussion of our findings in light of this literature has been added on lines 1031-1040:

“Recent investigations into the unitary LFP (the LFP generated by a single neuron) from inhibitory and excitatory synapses have suggested that inhibitory inputs exert a greater influence on LFP than excitatory input (Bazelot et al., 2010; Teleńczuk et al., 2017; B. Teleńczuk, M. Teleńczuk, and Destexhe, 2020). While this may be true of unitary effects, the total effect of excitatory input can still be greater if there are significantly more excitatory than inhibitory cells, and, correspondingly, significantly more excitatory synapses. In this V1 model, inhibitory cells make up about 15 % and excitatory cells about 85 % of the total number of cells, reflecting the cellular composition in mouse V1. Whether the pronounced dominance of excitatory cells is enough to make up for the reduced unitary influence of excitatory cells is an interesting area for further research.”

8. A more useful analysis to quantify the impact of specific network features would be a parameter sensitivity analysis that shows the influence of changes in parameters to variance in the resultant CSD; this isn't feasible for all parameters but could perhaps be done for the parameters/network features concluded to account (or not account) for the main feature of the canonical CSD, e.g. as summarized beginning on line 827. Or at least some simulations that show what happens over a range of parameter values for the important parameters.

This is a good suggestion, but the analysis in this paper is already extensive in scope. We therefore postpone such an analysis to a future paper.

9. A similar large scale biophysically detailed cortical column modeling paper (Reimann et al. Neuron 2013) is relevant to discuss further in light of the current results. In particular, that paper showed that "active currents" dominated the LFP, rather than synaptic currents. There is a statement line 890 that "In line with observations made by Reimann et al. (2013), we found that the somatic voltage- dependent membrane currents significantly shape the CSD signature (Figure 5H and 7B)". To my knowledge, Riemann considered the contribution of all active currents, including those in the dendrites, while in this paper they have only explored removing them in the soma. Those in the dendrites may have a greater impact, or maybe not (?). Can the authors show this? Further examination/discussion would be useful.

The number and level of detail of biophysically detailed neurons in this model unfortunately makes simulations with active conductances in the dendrites prohibitively expensive at this time.

10. Of particular relevance to these results are earlier laminar recordings to flash evoked responses in VI in non-human primates that suggest the presence of "feedback" and discusses interaction between higher and lower order areas that may contribute to this "feedback": doi: 10.1016/0042-6989(94)90156-2 https://doi.org/10.1093/cercor/8.7.575. A discussion on the conclusions in this paper related to those is needed.These papers, and many other papers, also show a decoupling between LFP/CSD and firing (MUA). Clarification should be made that such decoupling is not a new concept, but to my knowledge it has yet to be shown how it could be mechanistically implemented using a computational neural model, as the authors have nicely done. Nor has it been shown how simulated LFP/CSD can be altered with only minor effects on the firing rates, as shown here.Also relevant, is a more recent study of flash evoked responses in humans and non-human primates doi: 10.1126/sciadv.abb0977, which appears to show consistent results to those reported here.

Discussions of these papers have been added on lines 897-901 and 966-970. We see no explicit mention of feedback in these papers, but we do discuss whether CSD patterns in our data align with the patterns described in those papers as well as their observations on decoupling between LFP/CSD and MUA. Other studies by the same group that focus on the effect of feedback in evoked V1 responses are discussed on lines 974-979.

On lines 897-901:

“Studies in non-human primates where the animals were exposed to flashes also demonstrate good spatio-temporal agreement with the CSD observed here, with sinks and sources occurring not only in the same layers and in the same order, but also at the same time after stimulus presentation (Givre, Schroeder, and Arezzo, 1994; Schroeder, Mehta, and Givre, 1998).”

On lines 966-970:

“These results align with findings in e.g. Schroeder, Mehta, and Givre, 1998 and Leszczyński et al. 2020, where a lack of correlation between MUA and CSD in the upper layers of V1 during flash exposure to non-human primates suggested that these signals can sometimes be decoupled.”

On lines 974-979:

“Past studies have suggested that LFP can be modulated by attention through feedback during evoked responses (Mehta, Ulbert, and Schroeder, 2000a; Mehta, Ulbert, and Schroeder, 2000b). However, their findings indicated that V1 was not significantly affected by this modulation until the period 250 ms or later after stimulus onset. A recent study showed that feedback from higher visual areas can in fact exert a strong influence on the magnitude of LFP already at around 80 ms after stimulus onset (Hartmann et al., 2019).”

There were many interesting suggestions for further analyses in our study, and though we address many of them, as described above, several would, in our view, generate enough material to warrant another paper, which appears excessive. Our manuscript is already extensive and ambitious in scope and level of detail, and, we believe, has provided insights that no other study so far has been able to offer. We believe the revisions described above have substantially improved this work and address the most important concerns of the reviewers.